# Structural and dynamic insights into supra-physiological activation and allosteric modulation of a muscarinic acetylcholine receptor

Jun Xu [1,2,10], Qinggong Wang[3,4,10], Harald Hübner [5], Yunfei Hu[6,7], Xiaogang Niu[6], Haoqing Wang[1], Shoji Maeda [1,8], Asuka Inoue [9], Yuyong Tao [4], Peter Gmeiner [5], Yang Du [3]✉, Changwen Jin [6]✉ & Brian K. Kobilka [1]✉

The M2 muscarinic receptor (M2R) is a prototypical G-protein-coupled receptor (GPCR) that serves as a model system for understanding GPCR regulation by both orthosteric and allosteric ligands. Here, we investigate the mechanisms governing M2R signaling versatility using cryo-electron microscopy (cryo-EM) and NMR spectroscopy, focusing on the physiological agonist acetylcholine and a supra-physiological agonist iperoxo, as well as a positive allosteric modulator LY2119620. These studies reveal that acetylcholine stabilizes a more heterogeneous M2R-G-protein complex than iperoxo, where two conformers with distinctive G-protein orientations were determined. We find that LY2119620 increases the affinity for both agonists, but differentially modulates agonists efficacy in G-protein and β-arrestin pathways. Structural and spectroscopic analysis suggest that LY211620 stabilizes distinct intracellular conformational ensembles from agonist-bound M2R, which may enhance β-arrestin recruitment while impairing G-protein activation. These results highlight the role of conformational dynamics in the complex signaling behavior of GPCRs, and could facilitate design of better drugs.

G-protein-coupled receptors (GPCRs) comprise the largest family of transmembrane proteins and are remarkably versatile signaling molecules. GPCRs modulate cellular functions via coupling to heterotrimeric G-proteins or β-arrestins in a ligand-specific manner.

Ligands of a given GPCR can display different efficacy profiles, ranging from inverse agonists, to partial agonists and to full agonists[1,2]. In particular, there are biased ligands that can preferentially activate one signaling pathway over others[3,4], and allosteric modulators that can

[1]Department of Molecular and Cellular Physiology, Stanford University School of Medicine, Stanford, CA 94305, USA. [2]Beijing Advanced Innovation Center for Structural Biology, School of Medicine, Tsinghua University, 100084 Beijing, China. [3]Kobilka Institute of Innovative Drug Discovery, School of Life and Health Sciences, Chinese University of Hong Kong, 518172 Shenzhen, China. [4]Division of Life Sciences and Medicine, University of Science and Technology of China, 230027 Hefei, P. R. China. [5]Department of Chemistry and Pharmacy, Medicinal Chemistry, Friedrich-Alexander University, 91058 Erlangen, Germany. [6]Beijing Nuclear Magnetic Resonance Center, College of Chemistry and Molecular Engineering, Peking University, 100084 Beijing, China. [7]Innovation Academy for Precision Measurement Science and Technology, CAS, 430071 Wuhan, China. [8]Department of Pharmacology, Medical School, University of Michigan 1150 Medical Center Dr., 1315 Medical Science Research Bldg III, Ann Arbor, MI 48109, USA. [9]Graduate School of Pharmaceutical Sciences, Tohoku University, Sendai 980-8578, Japan. [10]These authors contributed equally: Jun Xu, Qinggong Wang. ✉e-mail: yangdu@cuhk.edu.cn; changwen@pku.edu.cn; kobilka@stanford.edu

bind to a site that is topographically distinct from the endogenous ligand binding site (orthosteric site)[5,6]. These ligands provide a means to fine-tune the biological responses to the natural signaling patterns and open new avenues for developing better therapeutic agents[7]. Consequently, it is of great interest to understand the molecular basis underlying GPCR regulation by orthosteric and allosteric ligands having different efficacy and signaling bias profiles.

Muscarinic acetylcholine receptors (mAChRs), particularly the M2 receptor (M2R), have long served as important model systems for understanding the mechanism of GPCR signaling[8]. These receptors are attractive drug targets for the treatment of a range of diseases, including Alzheimer's disease, schizophrenia, cardiac arrhythmias, and overactive bladder[9,10]. As such, a variety of drugs targeting muscarinic receptors have been developed over the past decades[11,12]. Of particular

interest is the synthesis of iperoxo (Ixo, Fig. 1a)[13], which was later found to be a supra-physiological agonist with higher efficacy profiles than the endogenous agonist acetylcholine (ACh, Fig. 1a) at the M2R, a phenomenon that is not commonly reported for GPCRs[14]. The super-agonism of Ixo facilitated the crystallization of M2R in its active state with the aid of a conformational selective nanobody (Nb9-8)[15]. Most recently, Ixo was also used to obtain the cryo-EM structures of the M2R-GoA and M2R-β-arrestin-1 signaling complexes[16,17]. These structures have significantly improved our understanding of agonist binding and activation of mAChRs, as well as the interaction interfaces with different signaling proteins. However, the mechanism underlying the super-agonism of Ixo remains poorly understood due to the lack of structural information on mAChRs bound to the physiological agonist ACh.

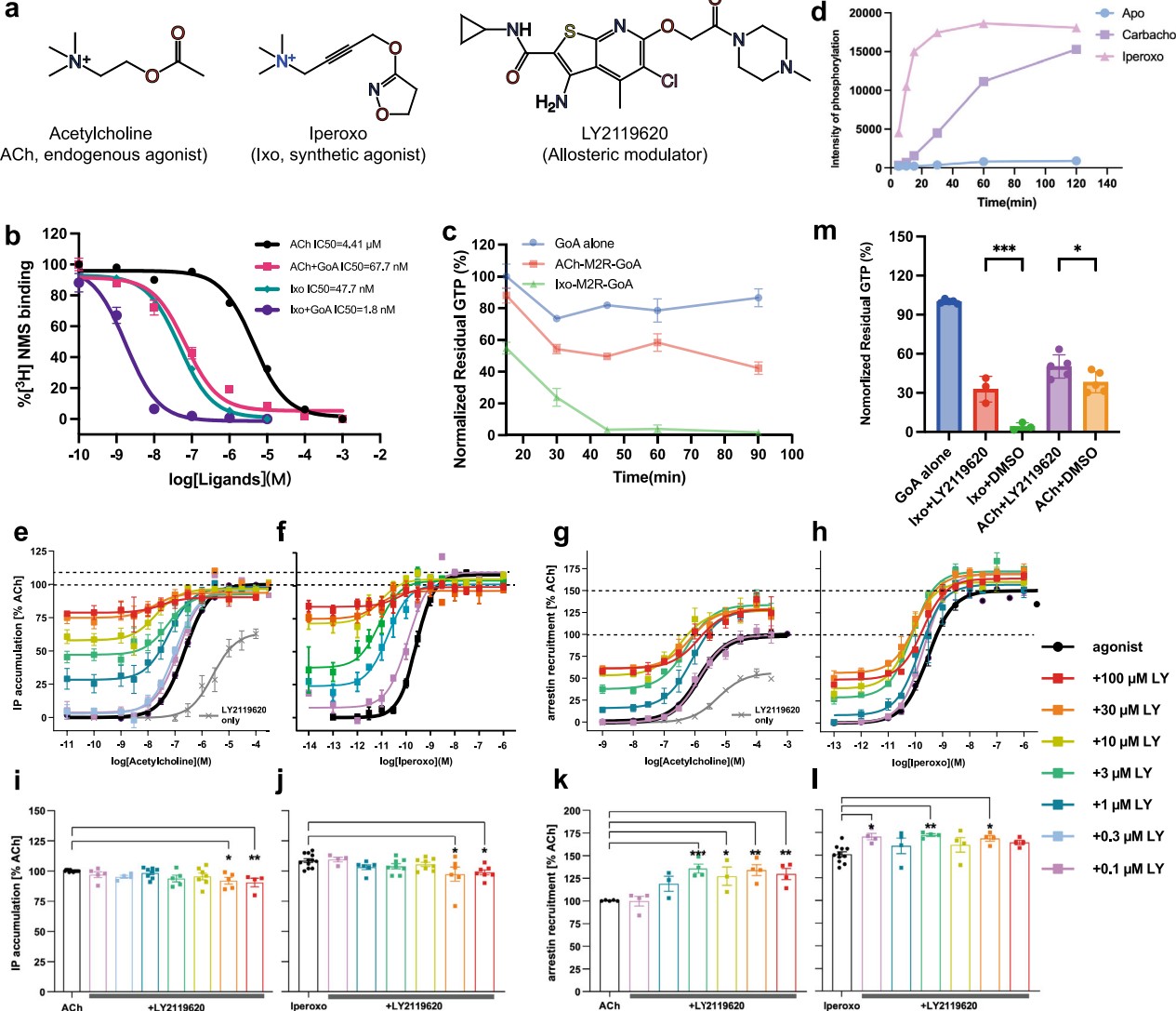

**Fig. 1 | Structures and function of M2R ligands. a** Chemical structures of ACh, Ixo, and LY2119620. **b** ACh and Ixo competition curves of the M2R reconstituted into HDL particles in the presence or absence of GoA. Data are given as mean ± SEM from three independent experiments. **c** GTP-turnover of M2R bound to ACh or Ixo at different time points. Data are given as mean ± SEM from three or four independent experiments. **d** Ligand-dependent phosphorylation of M2R by GRK2 as a function of time. Intensity were extracted using Fiji[75] from gel in Fig. S1B. **e**–**h** Concentration-response curves of ACh (**e**, **g**) and Ixo (**f**, **h**) toward G-protein activation and β-arrestin-2 recruitment in the presence of different concentrations of LY2119620. Dash lines indicate the maximal response of ACh and Ixo. Data with

error bars are presented as mean ± SEM of three to fourteen independent experiments. **i**–**l** Statistics of $E_{max}$ calculations for G-protein activation (**i** and **k**) and β-arrestin-2 recruitment (**j**, **l**). $E_{max}$ values were analyzed by One-way ANOVA applying two-sided Dunnett's multiple comparisons in PRISM 8.0. ***$p < 0.001$; **$p < 0.01$; *$p < 0.05$. **m** Effects of LY2119620 on G-protein activation efficacy of ACh and Ixo measured by GTPase Glo™ assay. Statistical analyses were performed using the ordinary one-way ANOVA followed by the two-sided Sidak's post-hoc test in PRISM 9.2.0. ***$p < 0.001$; *$p < 0.05$. Data with error bars are presented as mean ± SEM of three or five independent experiments.

The M2R is one of the best-characterized Family A GPCRs for understanding GPCR allostery[18–20]. A wide spectrum of allosteric modulators with varying pharmacological profiles have been characterized for the M2R[12]. LY2119620 (Fig. 1a) was the first positive allosteric modulator (PAM) to be co-crystallized with a mAChR[15], representing a major advance in understanding GPCR allostery at an atomic level. However, the active structures of M2R with or without the PAM are remarkably similar and these static structural snapshots do not fully explain the complex allosteric behavior of LY2119620.

In this work, we report the cryo-EM structures of the M2R-GoA signaling complex bound to ACh in the presence and absence of LY2119620. Moreover, we use NMR spectroscopy to investigate the conformational dynamics of the receptor and its transducer complexes regulated by ACh, Ixo, and LY2119620. We find that unlike Ixo, ACh cannot stabilize a uniform complex with nucleotide-free GoA. We also observe that GoA and β-arrestin stabilize distinct conformations at the intracellular surface of the M2R, and that LY2119620 can alter the intracellular conformational ensemble of Ixo- and ACh-bound M2R, which may lead to the enhancement of β-arrestin recruitment while suppressing G-protein activation for both agonists. Together, these studies provide important structural and dynamic information for understanding the molecular mechanisms of ligand recognition, activation, allostery as well as signaling bias of muscarinic receptors.

## Results

### Functional differences between ACh and Ixo and allosteric regulation by LY2119620

We first confirmed the functional differences between ACh and Ixo in vitro using purified receptors. Consistent with previous results from cell- or membrane-based assays, the binding affinity of ACh is much lower (-90 fold) than Ixo on purified M2R reconstituted in HDL particles. The addition of GoA can substantially increase the affinities of ACh and Ixo by -65 fold and -27 fold, respectively (Fig. 1b), suggesting both ACh- and Ixo-bound M2R can efficiently couple with G-protein. GTPase-Glo™ assay showed that ACh-bound M2R has a much lower GTP turnover rate and efficacy than Ixo-bound M2R (Fig. 1c). In addition, an in vitro phosphorylation assay shows that Ixo is more efficient than carbachol, an ACh analog (Supplementary Fig. 1a), to induce GRK2-mediated phosphorylation of M2R (Fig. 1d and Supplementary Fig. 1b). These results suggest that the supraphysiological effect of Ixo over ACh in both G-protein and β-arrestin signaling pathways.

Previous functional studies using [35S]-GTPγS-binding and ERK1/2 phosphorylation assays have shown that LY2119620 is a positive allosteric modulator of M2R which can dramatically enhance the affinity of both ACh and Ixo while having a negligible effect on the efficacy of orthosteric agonists[15,21]. As the ERK1/2 phosphorylation assay involves both G-protein and arrestin pathways[22,23], the effects of LY2119620 on arrestin signaling is still unclear. We then performed pathway-specific assays (G-protein IP-one accumulation and β-arrestin recruitment) to evaluate the effects of LY2119620 on signaling behaviors of ACh and Ixo under the same conditions (Fig. 1e–l and Supplementary Table 1). Consistent with in vitro studies, we observed higher efficacies in both G-protein activation and β-arrestin recruitment for Ixo compared to ACh (Fig. 1e–h). The potencies ($EC_{50}$) for both ACh and Ixo are significantly increased in the presence of LY2119620 (Fig. 1e–h and Supplementary Table 1), in agreement with previous functional data[21]. In contrast to the results from previous ERK1/2 phosphorylation assay[21], we observed significant enhancement of the efficacy ($E_{max}$) in β-arrestin recruitment for ACh (-30%) and Ixo (-15%) in the presence of high concentration of LY2119620, suggesting the positive allosteric effects of LY2119620 in the β-arrestin pathway (Fig. 1g, h, k, l and Supplementary Table 1).

Of interest, we found that high concentrations of LY2119620 could inhibit the maximal G-protein activation (-10%) for both orthosteric agonists, despite the fact that the modulator can activate the receptor on its own with low efficacy (Fig. 1e, f, i, j and Supplementary Table 1). Notably, this finding is in agreement with previous functional studies on LY2033298, a modulator with the same core scaffold as LY2119620 (Supplementary Fig. 1a), which also showed positive allosteric effects on ACh binding while the negative allosteric effect on ACh efficacy in [35S]-GTPγS-binding and ERK1/2 phosphorylation[24]. Because the engineered G-proteins used in these assays or the endogenous G-proteins in cell membrane could alter the functional outcomes, we further evaluated the allosteric effect of LY2119620 on G-protein activation using GTPase-Glo™ assay performed with purified wild-type receptor and G-protein. Consistent with the G-protein IP-one accumulation assay, we observed significant inhibitory effects of LY2119260 on GTP-turnover for both ACh and Ixo (Fig. 1m). Together, the functional analysis suggests that the modulator LY2119620 not only has positive allosteric effects on ACh and Ixo affinity, but also can modulate their signaling bias by enhancing β-arrestin recruitment while inhibiting G-protein activation. Of note, such pathway biased allosteric ligands have also been identified in several other GPCRs[25], for example, PDC113.824, an allosteric modulator of Prostaglandin F2α (PGF2α) receptor, shows positive effects in Gq-mediated signaling while shows negative effects in the G12-dependent pathway[26].

### Structural determination of M2R-GoA complex with ACh

To better understand the difference in efficacy between ACh and Ixo, we obtained the structure of the M2R-GoA complex with ACh using cryo-EM. We used the previously described ICL3 truncated construct (M2Rmini)[15] for our structural studies (Supplementary Fig. 1c). Despite the weaker potency and efficacy of ACh (Fig.1), ACh-bound M2R was able to form a biochemically stable complex with GoA in the presence of an antibody fragment scFv16[27], as shown by size exclusion chromatography and single particles in cryo-EM images (Supplementary Fig. 2a, b). The ACh-M2R-GoA complex was assembled using a similar strategy as described for the Ixo-M2R-GoA complex[16]. Unexpectedly, we observed two major conformers with distinct G-protein coupling orientations from the three-dimensional classifications of well-defined complexes (Supplementary Fig. 2c–e), and reconstructed the cryo-EM maps of the two states (denoted as S1 and S2) with overall resolution at 3.21 Å and 3.32 Å respectively (Fig. 2a, b, Supplementary Fig. 2f–i, and Table 2). As the previous Ixo-bound complex structure was determined in the presence of LY2119620, where only one conformer was resolved[16], we further examined whether LY2119620 can stabilize the ACh-bound signaling complex into a more uniform conformation. However, we observed the same two major conformers in the LY2119620-bound sample (Supplementary Fig. 2j–m), suggesting that the conformational heterogeneity at the G-protein coupling interface is dependent on the orthosteric ligand, but not the allosteric modulator. The cryo-EM maps of the two LY2119620-bound states were determined with overall resolution at 3.16 Å and 3.22 Å, respectively (Fig. 2c, d, Supplementary Fig. 2l–q, and Table 2). These high-resolution maps allowed us to build a model for most regions of the M2R, the GoA heterotrimer and the scFv16 (Fig. 2 and Supplementary Fig. 3a–d). Moreover, the densities for ACh are well-defined in these maps (Fig. 2). We also observed densities attributed to the bound-LY2119620 in the extracellular vestibule by comparing cryo-EM maps with and without the modulator (Supplementary Fig. 3e–h), although the density maps of the modulator are not well-defined due to the relatively low resolution at the extracellular surface (Supplementary Fig. 2p–q). Of interest, the resolution of the extracellular vestibule is lower for the structures bound to LY2119620 (Supplementary Fig. 2h, i, p, q). This may be related to the observation that LY2119620 is a negative allosteric modulator (NAM) for GoA efficacy, and the extracellular vestibule may be less compatible with LY2119620 binding in a GoA coupled receptor.

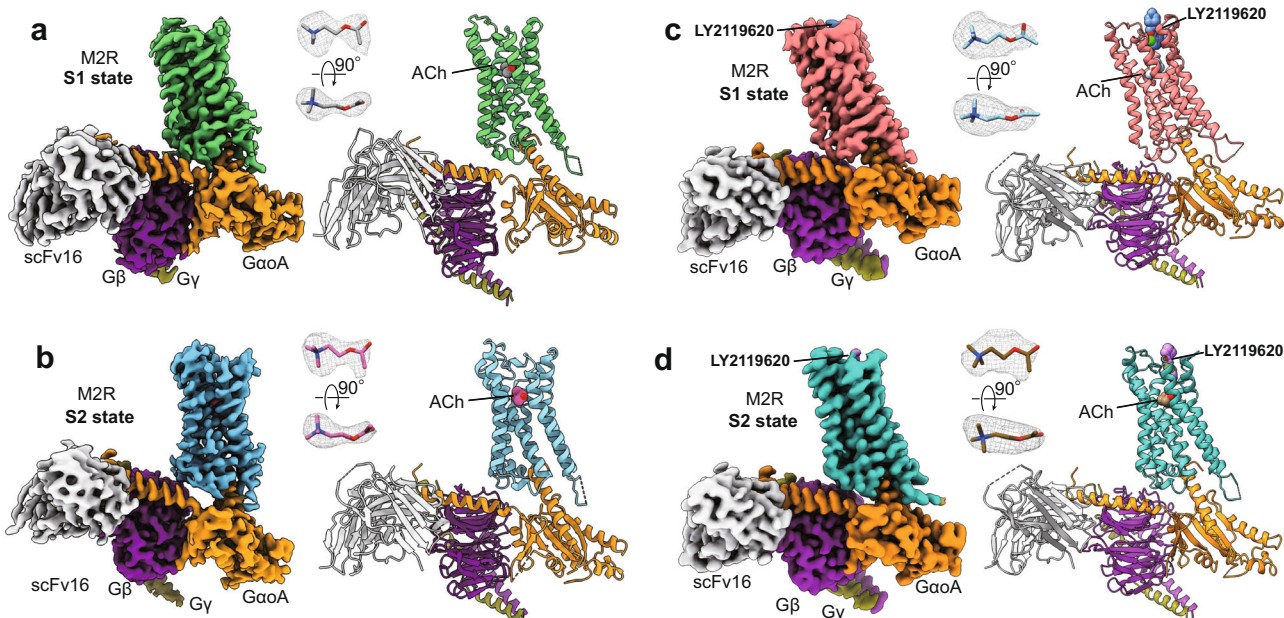

**Fig. 2 | Overall structures of M2R-GoA complexes. a, b** Cryo-EM density maps and models of the M2R-GoA complex bound to ACh in S1 (**a**) and S2 states (**b**). **c, d** Cryo-EM density maps and models of the M2R-GoA complex bound to ACh and the PAM LY2119620 in S1 (**c**) and S2 (**d**) states. The maps and models are colored according to polypeptide chains. The map densities of ACh (shown as sticks) are depicted as gray meshes.

## Comparison of M2R structures and orthosteric binding pocket

The overall structures of ACh-bound M2R in both S1 and S2 states are highly similar to the previous crystal structure of the Ixo-bound active M2R, with root mean square deviations (r.m.s.d) of 0.818 Å and 0.982 Å, respectively (Fig. 3a). The rotamers of conserved motifs or residues, including the $P^{5.50}$-$V^{3.40}$-$F^{6.44}$ core triad, $Y^{5.58}$, $Y^{7.53}$ (NPxxY motif) and $R^{3.50}$ (DRY motif), are almost identical with those of the Ixo-bound active structure, suggesting that both ACh-bound S1 and S2 represent fully active M2R (Fig. 3b, c). Notably, we observed small differences in the ligand binding domain between the two states. As shown in Fig. 3d, ACh binds to the deep orthosteric pocket constituted by TM3, 5, 6, and 7. The densities of the orthosteric residues are all well-defined in both states (Fig. 3e, f). $Y104^{3.33}$, $Y403^{6.51}$, and $Y426^{7.39}$ constitute an aromatic tyrosine lid on the top of ACh and form cation-π interactions with the amine moiety. The residue $D103^{3.32}$, which is highly conserved among the aminergic GPCRs, engages in an electrostatic interaction with the trimethyl ammonium ion of ACh. We also resolved a probable water molecule in the S2 state, which is positioned to mediate a hydrogen-bonding interaction between ACh and $N404^{6.52}$ (Fig. 3d, f), consistent with previous vibrational spectroscopic[28] and molecular dynamics (MD) simulation studies[29], as well as functional studies that show $N404^{6.42}$ plays a key role in ACh binding[15]. Of interest, the tyrosine lid shows different sidechain conformations between S1 and S2 states, where $Y426^{7.39}$ undergoes a ~70° rotation toward the extracellular side in S2 state, resulting in slightly open lid conformation (Fig. 3d–f). A similar rotamer conformation of the homologous $Y^{7.39}$ was observed in the apo-state M4R structure (Supplementary Fig. 3i)[30], suggesting that the conformational heterogeneity of $Y^{7.39}$ may be conserved within muscarinic receptors and functionally important. In addition to $Y426^{7.39}$, we also observed two alternative sidechain rotamers for $W427^{7.40}$ in the S2 state, along with slight differences in other aromatic residues (e.g. $W422^{7.35}$ and $Y80^{2.61}$) clustered at the interface of TM2 and TM7 (Fig. 3g and Supplementary Fig. 3j). Together, these structural observations suggest that ACh fails to stabilize a uniform conformation in the extracellular domain even in the presence of intracellular G-protein. We assumed that the different conformations in the ligand binding domain may represent different affinity states of ACh within the receptor-G-protein complex, which is likely associated with the

different intracellular G-protein coupling modes. Indeed, numerous biochemical and biophysical studies have shown that G-protein coupling can allosterically modulate the conformation of the orthosteric pocket of GPCRs[31,32]. In addition, previous functional studies also showed heterogeneous Kd values of the high-affinity (G-protein coupled) state of ACh depending on the G-protein concentration[33].

In contrast to ACh-bound structures, Ixo-bound M2R structures have a relatively uniform conformation in the orthosteric pocket (Supplementary Fig. 3k). These observations are consistent with previous MD simulation studies, which showed that binding of Ixo leads to significantly decreased structural fluctuations in the orthosteric pocket of M2R bound to a G-protein mimetic nanobody compared to that of arecoline, a low potency agonist similar to ACh[34]. The Ixo-bound orthosteric conformation is more similar to the ACh-bound S1 state than the S2 (Fig. 3h). The ammonium moieties of Ixo and ACh-S1 essentially overlap and form similar interactions with the tyrosine lid and $D103^{3.32}$. However, due to the longer tail and bulkier isoxazoline ring, Ixo binds deeper into the receptor core and has more contacts with TM5 and 6, including van der Waals interactions with $A194^{5.46}$ and $F195^{5.47}$ and a direct hydrogen-bonding interaction with $N404^{6.52}$ (Fig. 3h). These additional interactions between Ixo and M2R may help stabilize a more uniform active conformation of the extracellular domain and lead to the higher binding affinity than that of ACh (Fig. 1b). Moreover, the conserved toggle switch $W400^{6.48}$ undergoes a rotation in Ixo-bound state while it remains an inactive-like conformation in ACh-bound state (Fig. 3h, i), suggesting that rotation of $W400^{6.48}$ is not necessary for the activation of M2R, but might contribute to the greater efficacy of Ixo (Fig. 1). Of interest, these observations are consistent with our previous dynamics studies using NMR spectroscopy and MD simulations[29], providing direct structural explanations for the distinct functional properties of ACh and Ixo.

## Comparison of G-protein coupling interfaces

To examine whether the different interactions with the M2R orthosteric pocket for ACh and Ixo could lead to different coupling modes with the G-protein, we next analyzed the receptor-GoA interface. Recent cryo-EM structures of GPCR-G-protein complexes have

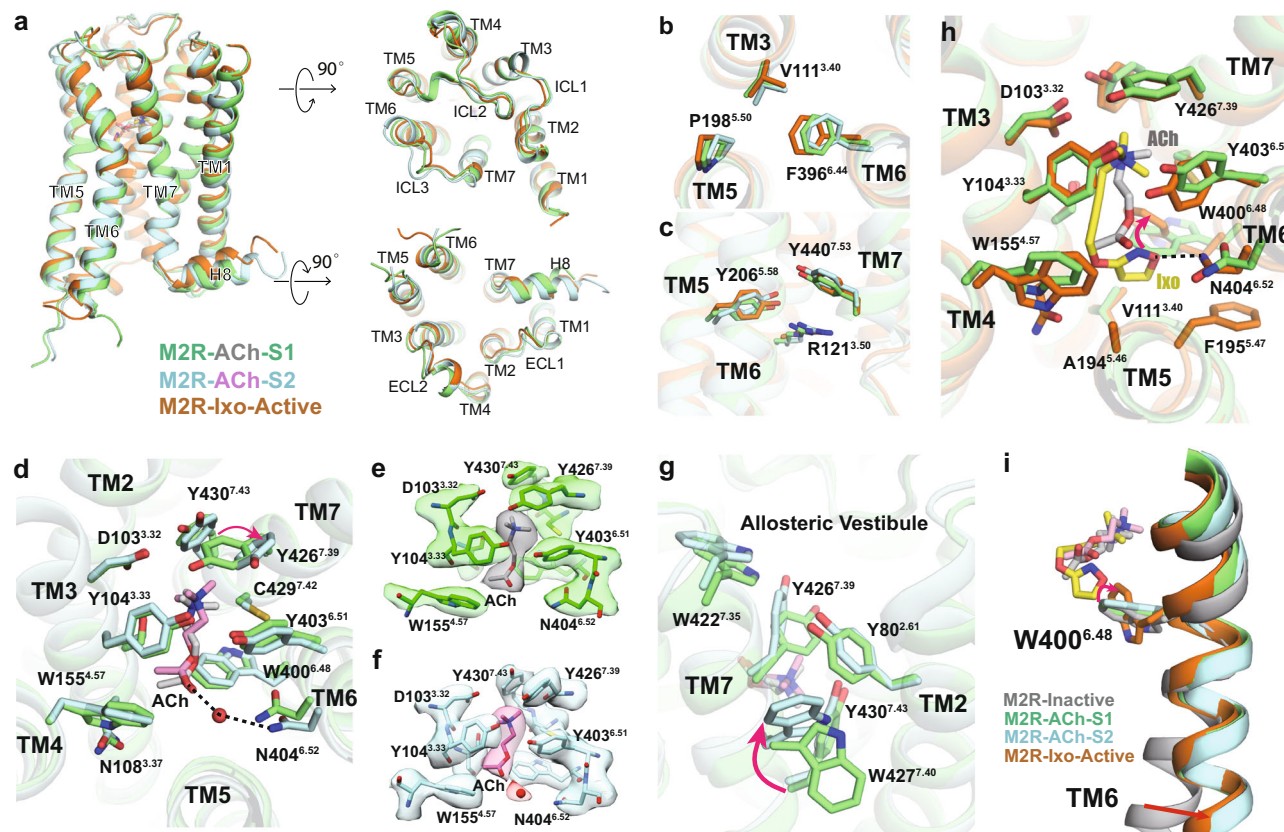

**Fig. 3 | Comparison of M2R structures and othosteric pocket. a** Comparison of overall M2R structures of ACh-bound S1 (green), ACh-bound S2 (cyan), and Ixo-bound (orange, PDB: 4MQS) active states from the orthogonal, extracellular, and intracellular views. **b, c** Comparison of the P5.50-V3.40-F6.44 core triad (**b**) and conserved side-chains R121³·⁵⁰, Y206⁵·⁵⁸, and Y440⁷·⁵³(**c**) in three different active conformations. **d** Comparison of ACh binding pockets in S1 and S2 states. Red arrow indicates the conformational differences for Y426⁷·³⁹. Black dashed lines indicate hydrogen-bond interactions with N404⁶·⁵². **e, f** Density maps of residues in the orthosteric pocket in S1 (**e**) and S2 (**f**) states. The probable water molecule in S2 state is shown as red sphere. **g** Comparison of the TM2-TM7 interface between S1 and S2 state. Red arrow indicates the conformational differences for W427⁷·⁴⁰. **h** Comparison of the Ixo and ACh binding pockets in S1 state. Red arrow indicates different rotamers of W400⁶·⁴⁸. **i** Comparison of the toggle switch W400⁶·⁴⁸ conformations when bound to antagonist (PDB: 3UON), ACh, and Ixo.

revealed variable binding modes of the G-protein, especially for the Gi/o family³⁵,³⁶. Although many complex structures have been determined in the presence of different agonists and/or allosteric ligands with distinct chemotypes or functional profiles³⁷⁻³⁹, few of them revealed different G-protein coupling modes for the same receptor³⁶,⁴⁰,⁴¹. Among them, the neurotensin receptor 1 (NTSR1)-Gi1 complex showed the most notable differences between two states, the canonical and non-canonical states (C and NC states), where the G-protein in the NC state is rotated by approximately 45° with a 25° tilt of the α5 helix compared to the C state³⁶,⁴². While the two states show identical conformations in the ligand binding domain, the NC state NTSR1 displays an active conformation for the cytoplasmic end of TM6, but an inactive conformation for the cytoplasmic end of TM7, as such, the NC state was proposed to be an intermediate along the activation pathway of NTSR1³⁶. In contrast, the intracellular side of M2R for both of the M2R-GoA complexes displays similar fully active conformations (Fig. 3b, c), despite the differences in G-protein orientation. Therefore, we are unable to assign the sequence of the two states along the signaling pathway of the M2R. Comparison with the NTSR1-Gi complex structures shows that the G-protein coupling modes in both S1 and S2 are more similar to the C state of NTSR1-Gi and the G-proteins are all in nucleotide-free state (Fig. 4a), suggesting that the two conformations of the M2R-GoA complex may represent two different active states in equilibrium, rather than two sequential events in the process of complex formation.

As the LY2119620-induced differences at the G-protein coupling interface for both states are negligible (Supplementary Fig. 4a), we used the structures without LY2119620 for the following comparison between S1 and S2 states. The GoA in the S2 state rotates around 30° relative to the receptor, along with a 10° tilt of the α5 helix compared to that of S1 state (Fig. 4b). While most of the residues that mediate the interactions between α5 helix and the receptor core remain unchanged between S1 and S2 states (Fig. 4c, d), several differences were observed. First, due to the slightly upward shift of the α5 helix in S2 state (Fig. 4b), we observed additional polar interactions between R121³·⁵⁰ and backbone of C351^{Gαo}, and between N444^{8.47} and the backbone of R349/G350^{Gαo} (Fig. 4c, d). Of note, the polar interaction between R³·⁵⁰ and C351^{Gαo} was also found in several other GPCR-Gi/o complex structures⁴³,⁴⁴. Second, rotation of GoA also dramatically alters the interaction interface between ICL2 and the hydrophobic patch formed by the α5 and αN helices of Gαo (Fig. 4e–g). The hydrophobic contacts are much stronger in the S1 state than those in the S2 state, mainly mediated by the conserved ICL2 residue L129³⁴·⁵¹ (Fig. 4e–g). Replacement of L129³⁴·⁵¹ with alanine significantly reduced the potency (pEC₅₀) for both ACh and Ixo, suggesting the important role of this residue in the full activity of GoA coupling (Supplementary Fig. 5). The interface between ICL1 and the Gβ subunit is larger in the S2 state than that of S1 state, and there are potential polar interactions between R52 in ICL1 and Gβ in S2 state (Supplementary Fig. 4b). Of note, direct hydrogen-bonding interactions between ICL1 and Gβ subunit were also found in several other GPCR-G-protein complexes, such as the α2BAR-Go and GPR97-Go complexes⁴⁴,⁴⁵. Together, these results suggest the dynamic nature for the coupling between ACh-bound M2R and GoA, where most of the core interactions remain the

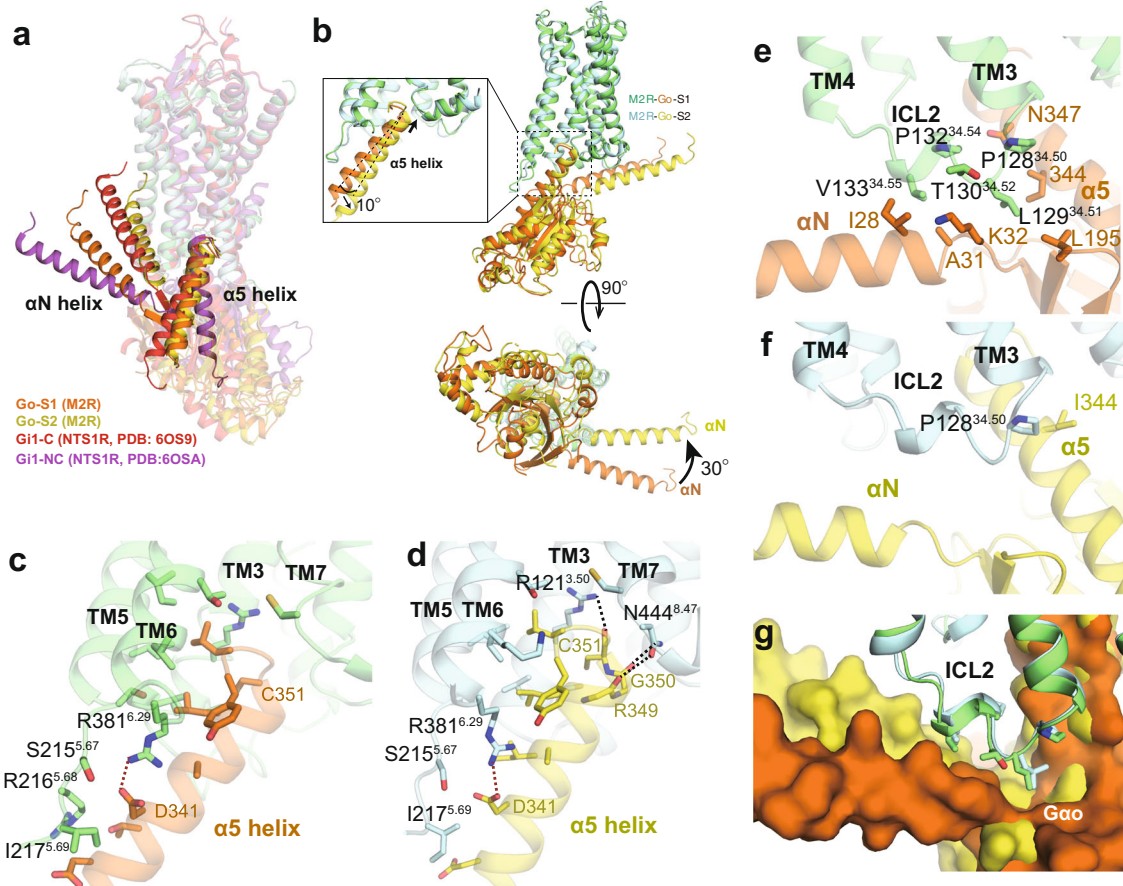

**Fig. 4 | Comparison of G-protein coupling interface between S1 and S2 states. a** Comparison of G-protein orientations between M2R-GoA and NTS1R-Gi1. PDB: NTS1R-Gi1 C state (6OS9), NC state (6OSA). **b** Side (up) and intracellular (bottom) views of the superposed structures of M2R-GoA complex in S1 and S2 states. **c**, **d** Interaction between α5 helix of Gαo and M2R in S1 (**c**) and S2 (**d**) states. Dashed lines represent polar interactions. **e**, **f** Interaction between α5 and αN helices of Gαo and ICL2 of M2R in S1 (**e**) and S2 (**f**) states. **g** Surface view of the hydrophobic pocket in Gαo that interacts with ICL2 of M2R.

same while the interaction interfaces between intracellular loops and G-protein vary between different states.

## ACh and Ixo stabilize different conformational dynamics within M2R-GoA signaling complex

In contrast to the conformational heterogeneity of the ACh-stabilized M2R-GoA complex, cryo-EM analysis of the Ixo-bound M2R-GoA complex revealed only one conformation[16]. Interestingly, structural comparison shows that the G-protein orientation in Ixo-M2R-GoA complex is different from both states of ACh-M2R-GoA (Fig. 5a). The αN helix of Ixo-M2R-GoA is more similar to that of S2 state than the S1 state (Fig. 5b), while the α5 helix of Ixo-M2R-GoA is positioned between the S1 and S2 states (Fig. 5c). We speculate that Ixo also stabilizes different sub-states with subtle differences such that an averaged conformation was resolved by cryo-EM. Indeed, recent 3D variability analysis (3DVA) resolved high resolution continuous flexibility within the 3 Å cryo-EM structure of CB1-Gi1 signaling complex[46,47]. These observations further suggest the existence of diverse coupling modes between M2R and GoA and that ACh stabilizes different conformational dynamics of the signaling complex from that of the supra-physiological Ixo.

As described earlier, we found that the major differences in the orthosteric pocket between Ixo and ACh are their interactions with TM6 (Fig. 3h, i). It is well accepted that TM6 movement is a hallmark of class A GPCR activation. A number of biophysical studies have shown that TM6 conformational dynamics is correlated with downstream signaling efficacies[48,49]. Using $^{13}CH_3$-ε-methionine ($^{13}CH_3$-ε-Met) NMR,

we previously investigated the conformational dynamics of the M2R bound to distinct orthosteric modulators, revealing the conformational plasticity of M2R, where we found that ACh and Ixo stabilize remarkably different states in the transmembrane domain of M2R[29]. To further examine whether such differences could cause different TM6 dynamics within the M2R-GoA complex, which in turn lead to distinct G-protein coupling modes, we engineered a TM6 $^{13}CH_3$-ε-Met probe by introducing a K383[6.31]M mutation at the cytoplasmic end of TM6 and monitored its conformational states in the process of complex formation using NMR spectroscopy (Fig. 5d and Supplementary Fig. 1c). Radio-ligand binding studies show that the engineered construct show similar ligand binding and G-protein coupling properties to the wild type (WT) M2R (Supplementary Fig. 4c, d). The resonance of K383[6.31]M was assigned at around 2.16 ppm in $^{13}C$ dimension and 17.2 ppm in $^{1}H$ dimension (Supplementary Fig. 4e). When bound to ACh or Ixo, similar spectral changes were observed, where two different conformational states were detected (Fig. 5e–g). When GDP-bound GoA is added, the spectrum of Met383[6.31] become more heterogeneous, suggesting the existence of multiple conformational states of TM6, which likely arose from the unstable complex in the presence of GDP. Of interest, the ACh-M2R-GoA[GDP] bound spectrum is different from that of Ixo-M2R-GoA[GDP] bound spectrum, suggesting a different active conformation, or conformational ensemble, between ACh-bound and Ixo-bound M2R in the presence of GoA[GDP] (Fig. 5g, middle panel). Upon digesting the GDP with apyrase, the Met383[6.31] peak becomes more regular in Ixo-M2R-GoA[apyrase] bound condition, which may represent the formation of a more uniform nucleotide-free state as observed in

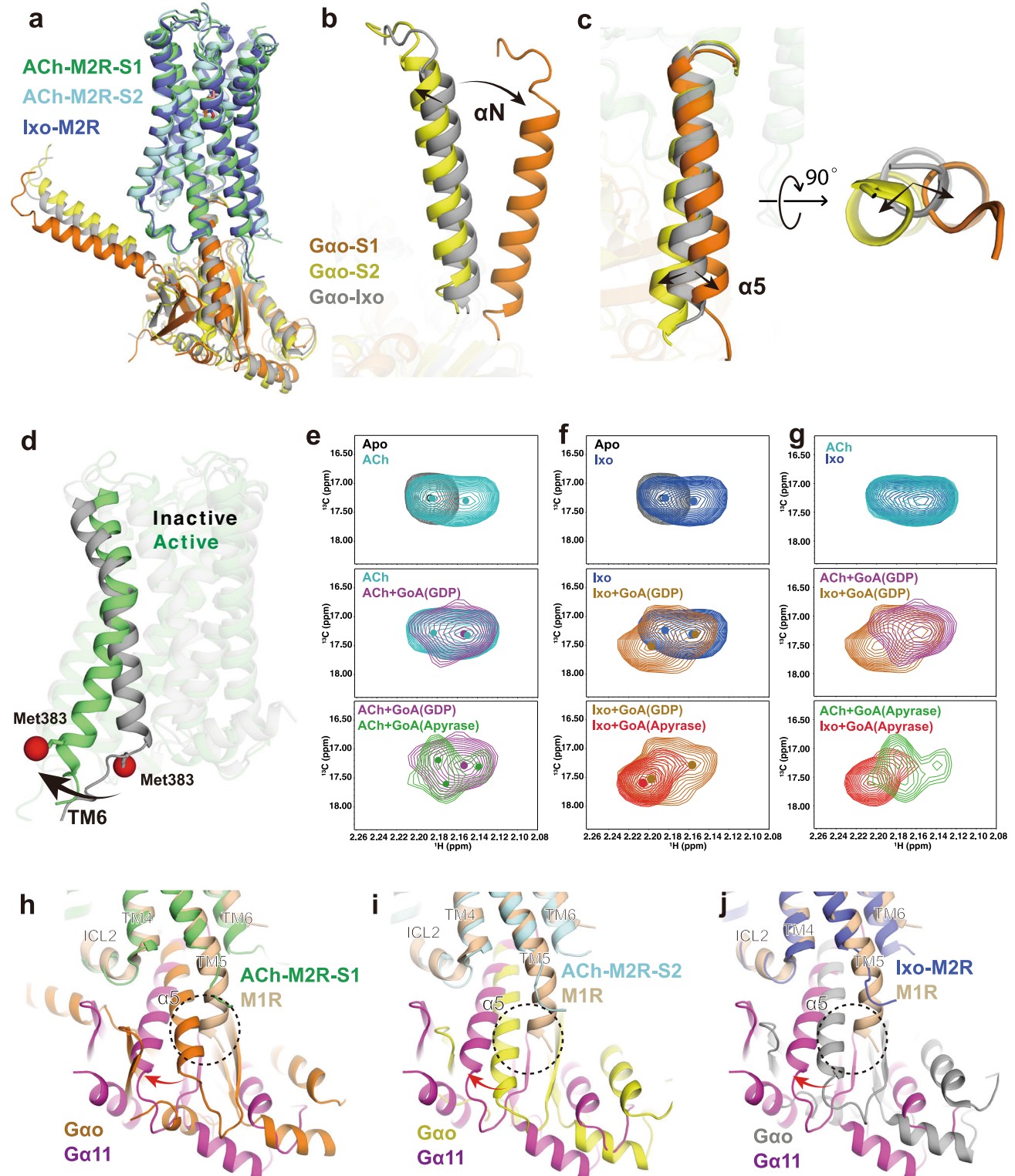

**Fig. 5 | Comparison of ACh and Ixo stabilized G-protein complex. a** Comparison of the orientation of Gαo in ACh-bound S1, ACh-bound S2, and Ixo-bound (PDB: 6OIK) states. **b**, **c** Comparison of αN and α5 helices in three structures, the α5 helix is shown in both side and bottom views. **d** Cartoon representation of the active (PDB: 6OIK) and inactive (PDB: 3UON) structures of the M2R with TM6 and the position of Met383[6.31] (red) highlighted. **e** HSQC spectra of Met383[6.31] in ACh-bound (cyan), ACh-GoA[GDP]-bound (purple), and ACh-GoA[apyrase]-bound (green) states. **f** HSQC spectra of Met383[6.31] in Ixo-bound (blue), Ixo-GoA[GDP]-bound (brown), and Ixo-GoA[apyrase]-bound (red) states. The apo-state spectrum of Met383[6.31] is shown in gray as a reference. **g** Comparison of the HSQC spectra of Met383[6.31] bound to ACh and Ixo along the process of GoA complex formation. The peak centers in ligand-bound spectra are shown as colored dots. **h**–**j** Alignment of the M1-G11 structure (PDB: 6OIJ) with the ACh-bound M2R-GoA S1 (**h**) and S2 (**i**) structures and the Ixo-bound M2R-GoA (PDB: 6OIK) structure (**j**). The different orientation between GoA and G11 is depicted with curved arrows. The clash between the extended TM5 helix with GoA is highlighted with dashed circle.

the cryo-EM structure. In the ACh-M2R-GoA[apyrase] bound state, the spectrum of Met383[6.31] also undergoes further changes, but we still observe multiple peaks, suggesting substantial conformational heterogeneity with a relatively slow exchange rate. Moreover, the overall peak center of Met383[6.31] in the ACh-M2R-GoA[apyrase] complex is different from that in the Ixo-M2R-GoA[apyrase] complex, suggesting different active conformers of the signaling complex stabilized by ACh and Ixo (Fig. 5g). These results agree with the heterogeneity observed in the cryo-EM structures. Taken together, these NMR studies reveal different conformational dynamics within the G-protein signaling complexes stabilized by ACh and Ixo, and further support the diverse coupling modes between M2R and GoA. These results are consistent with recent [19]F-NMR and single-molecule FRET studies showing that partial and full agonists stabilize different active conformers within the A2A-Gs and β2AR-Gs signaling complexes[48,50]. Moreover, we observed slightly different conformations for the β6-α5 loop of GoA between ACh- and Ixo-bound complexes (Supplementary Fig. 4f), which may directly contribute to the different GTP turnover rates (Fig. 1c). This is consistent with previous biochemical and functional studies on the α2A adrenergic receptor and calcitonin receptor that showed ligand-dependent modulation of G-protein conformational dynamics, which in turn alters drug efficacy[51,52].

Comparison of the ACh-bound signaling complex structures with the M1R-G11 structure also provides additional insights in the G-protein coupling specificity of muscarinic receptors (Fig. 5h–j). Previous structural studies revealed that one of the most distinct difference between M2R-GoA and M1R-G11 structures is the extended helical turns of TM5 in M1R-G11[16]. Interestingly, we found that this extended TM5 helix overlaps the position of the α5 helix of Gαo in the ACh-bound S1 state (Fig. 5h), while this direct steric clash is not obvious for the α5 helix in ACh-bound S2 state or Ixo-bound state (Fig. 5i, j). Together, these observations suggest that the dynamic coupling modes between receptor and G-protein could play an important role in coupling specificity.

## Insights into allosteric regulation of LY2119620 on agonist binding

In structures of M2R determined by crystallography with Ixo and Nb9-8, the active M2R with or without LY2119620 are remarkably similar, except for the distinct side chain of W422[7.35] in the extracellular vestibule (ECV), which has direct interactions with the PAM[15]. However, the difference is not sufficient to explain the complex allosteric effects of LY2119620 (Fig. 1). We then analyzed the ACh-bound signaling complex structures with and without LY2119620. Similar to previous studies, the modulator binds to the same allosteric site located in the extracellular surface and further stabilizes the contracted active conformation of the ECV (Fig. 6a, c). Notably, we observed more obvious conformational rearrangements in the ACh-bound receptor-G-protein complex upon binding with LY2119620. In the S1 state, the structure changes mainly occur in the ECV (Fig. 6a), while subtle differences were observed in the orthosteric pocket (Fig. 6b and Supplementary Fig. 6a). In addition to the rotamer changes for W422[7.35], we found conformational rearrangements for F181[45.55] in ECL2, which rotated approximately 180° (Fig. 6a). The density map of F181[45.55] was not well-resolved in previous LY2119620-bound structures due to relatively low resolution or the possibility of conformational flexibility[15,16]. Of note, F181[45.55] in M2R is different from the other four muscarinic subtypes (M1, M3-M5), which is leucine in the homologous position[53]. The water molecule-mediated polar interactions between ACh and N404[6.52] are found in the S1 state only when LY2119620 is bound, and in the S2 state only in the absence of LY2119620 (Fig. 3f and Supplementary Fig. 6a). It is likely that the water-mediated interaction exists in all conformations; however, the density was not well resolved in two of the maps. In the S2 state, LY2119620 leads to conformational changes in both the ECV and the orthosteric pocket. The changes in the ECV of the S2 state are

similar to those observed in the S1 state (Fig. 6c), while the orthosteric pocket of the S2 state with LY2119620 becomes more similar to that of the S1 state in the presence of the PAM, where Y426[7.39] rotates toward the ACh ammonium head, resulting in a more closed lid conformation and a slight inward movement of N404[6.52] (Fig. 6d and Supplementary Fig. 6b). Additionally, W427[7.40] has a more uniform conformation similar to that of the S1 state upon LY2119620 binding, together with slight changes of Y80[2.61] and Y83[2.64] in TM2 (Fig. 6e and Supplementary Fig. 3j). These results suggest that binding of LY2119620 alters the conformational dynamics of the extracellular domain, particularly for the aromatic network linking the allosteric and orthosteric sites (Fig. 6e).

Because M406[6.54] and M77[2.58] are ideally positioned to detect the structural changes of surrounding aromatic residues (i.e. W422[7.35] and W427[7.40]) in the extracellular domain (Fig. 6e), we further investigated the effects of LY2119620 on M2R conformational dynamics in the absence of intracellular proteins utilizing ε-[13]CH[3]-Met NMR. A modified M2R(M2Rmini△5 M) (Supplementary Fig. 1c) was used for better spectral quality of the extracellular probes[29]. The overall spectra of the M2Rmini△5 M in different conditions are summarized in Supplementary Fig. 6c–h. We found that LY2119620 alone stabilizes a different conformation relative to the apo-state receptor (Supplementary Fig. 6c, d), indicating that LY2119620 is able to bind to the receptor even without an agonist. Consistent with cryo-EM observations (Fig. 6c–e), we observed dramatic spectral changes in M406[6.54] and M77[2.58] upon adding LY2119620 to ACh-bound receptor, and similar effects were observed for Ixo-bound receptor (Fig. 6f, g). Collectively, these observations from cryo-EM structures and NMR spectra suggest that LY2119620 not only can stabilize the contracted active ECV conformation but also can regulate the conformational dynamics of the aromatic network linking the allosteric and orthosteric sites, which could provide structural explanations for the positive allosteric effects of LY2119620 on binding of ACh and Ixo (Fig. 1e, h). In agreement, previous mutagenesis studies at the homologous positions of the aromatic network in M4R (including Y[7.39], W[7.40], Y80[2.61], and Y83[2.64]) showed that replacement of these residues with alanine can significantly decrease the PAM cooperativity[54].

## Insights into allosteric regulation of LY2119620 on agonist signaling bias

In contrast to the extracellular ligand-binding domain, the conformations in the intracellular G-protein binding domain of ACh-bound M2R with or without LY2119620 are very similar in our cryo-EM structures (Fig. 7a, b). Notably, upon LY2119620 binding, obvious spectral changes were detected by NMR spectroscopy for the [13]CH[3]-ε-Met probe M202[5.54], which is located between the P[5.50]-I/V[3.40]-F[6.44] motif and cytoplasmic end of TM5, with the ε-[13]CH[3] group positioned near TM5, TM6, and TM7 (Fig. 7a). We find that the peak corresponding to M202[5.54] appears to be more heterogeneous than the apo-state peak (Fig. 7c, d, top panel), indicating that LY2119620 binding might be able to alter the dynamic behavior of the TM5-TM6 interface on the intracellular side, possibly by slowing down the chemical exchange between different states or by stabilizing different conformations. These spectral changes may reflect structural changes responsible for the allosteric partial agonism of LY2119620 (Fig. 1e, g).

When adding LY2119620 to ACh-bound or Ixo-bound M2R, a new M202[5.54] peak is observed between the peaks observed for apo M2R and M2R bound to agonist alone (Fig. 7c, d), suggesting the formation of a distinct intermediate active state. Additionally, a smaller peak is observed at a similar position to the peak in the apo-state, suggesting that for agonist plus LY2119620, the receptor undergoes slow conformational exchange between an inactive-like state and an intermediate active state, where the intermediate active state is different from that of M2R bound to agonist alone. However, we did not see obvious structural difference surrounding M202[5.54] in G-protein

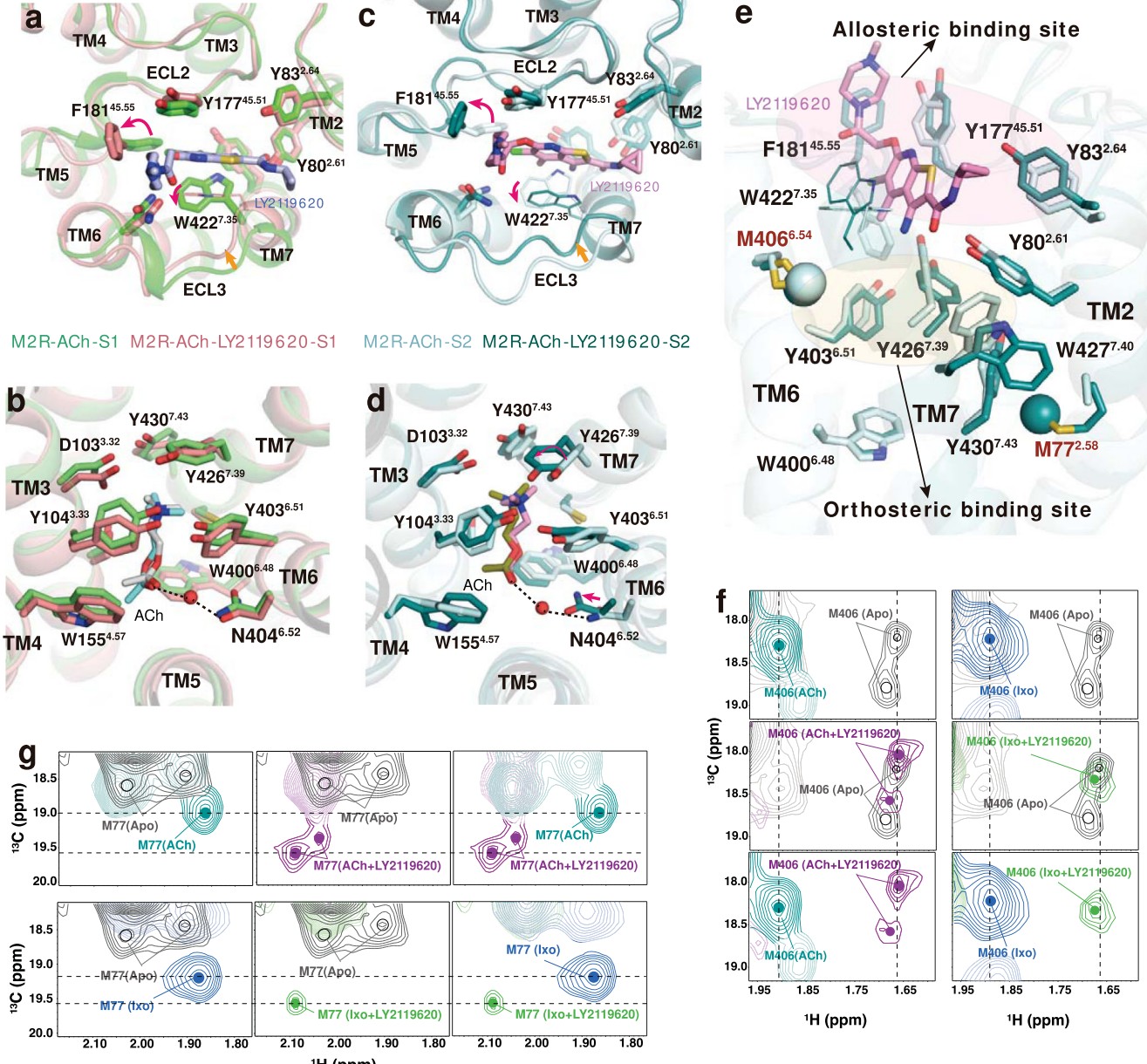

**Fig. 6 | Effects of LY2119620 on the M2R extracellular conformational changes.** **a**, **b** Structural comparison of the extracellular vestibule (**a**) and ACh binding pocket (**b**) in S1 state with or without LY2119620. **c**, **d** Structural comparison of the extracellular vestibule (**c**) and ACh binding pocket (**d**) in S2 state with or without LY2119620. **e** Structural comparison of the extracellular aromatic network in S2 state with or without LY2119620. The ε-$^{13}$CH$_3$ of M77$^{2.58}$ and M406$^{6.54}$ are shown as spheres. W422$^{7.35}$ shown in lines indicates the side chain doesn't robustly fit cryo-EM density. **f**, **g** The HSQC spectra of M406$^{6.54}$ (**f**) and M77$^{2.58}$ (**g**) in apo, ACh-bound, ACh/LY2119620-bound, Ixo-bound and Ixo/LY2119620-bound states. Color code: gray (apo), light teal (ACh), magenta (ACh/LY2119620), blue (Ixo), bright green (Ixo/LY2119620).

coupled cryo-EM structures with or without LY2119620 (Fig. 7a). It is possible that, in the M2R-GoA complex, the conformation of the cytoplasmic surface is dominated by interactions with GoA. Similarly, the nanobody stabilized active crystal structures with and without LY2119620 also showed the same intracellular conformation[15]. As M202$^{5.54}$ serves as an excellent probe for monitoring the conformational changes of the TM5/6/7 intracellular interface (Fig. 7a), the down-field shift of M202$^{5.54}$ toward the apo-state spectrum suggest that LY2119620 plus agonist may stabilize a more occluded active conformation of the TM5/6/7 intracellular cavity as compared to that of agonist alone, which could impair the coupling of G-protein. These data are in line with the functional data that show high concentrations of LY2119620 have inhibitory effect on G-protein signaling efficacy of ACh and Ixo (Fig. 1). Of interest, recent DEER spectroscopic studies on

the angiotensin II type 1 receptor (AT1R) and NMR studies on the μ-opioid receptor (μOR) revealed that β-arrestin-biased agonists or mutants favor more occluded intracellular conformations (smaller TM5/6/7 cavity) as compare to those G-protein biased or balanced agonists[55,56]. MD simulations studies of the AT1R also revealed an alternative active conformation in the intracellular domain of the AT1R which couples preferentially to arrestins[57]. We therefore speculated that the distinct conformational state stabilized by LY2119620 plus agonist (Fig. 7c, d) is biased for coupling with a GPCR kinase (GRK) and/or β-arrestin. This is supported by the markedly enhanced efficacy (E$_{max}$) in β-arrestin recruitment for ACh and Ixo in the presence of high concentrations of LY2119620 (Fig. 1). Moreover, recent cryo-EM structure of M2R-β-arrestin-1 complex displayed different local conformation surrounding M202$^{5.54}$, especially the Y440$^{7.53}$ in TM7, which

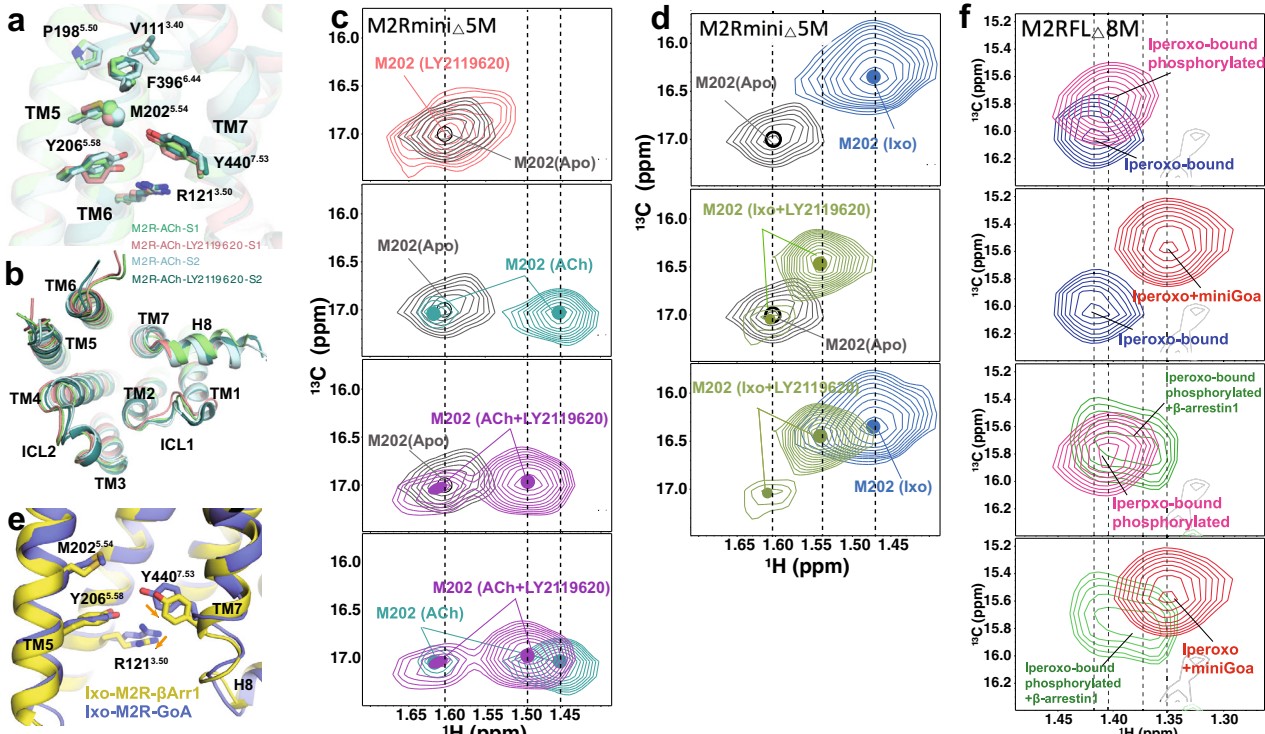

**Fig. 7 | Effects of LY2119620 on the M2R intracellular conformational changes.**
**a** Comparison of the $P^{5.50}$-$V^{3.40}$-$F^{6.44}$ core triad and conserved side-chains $R121^{3.50}$, $Y206^{5.58}$ and $Y440^{7.53}$ in the intracellular domain of S1 and S2 states with or without LY2119620. The $\epsilon$-$^{13}CH_3$ of $M202^{5.54}$ is shown as a sphere. **b** Comparison of the intracellular surfaces. **c**, **d** The HSQC spectra of $M202^{5.54}$ in apo, ACh-bound and ACh/LY2119620-bound, as well as Ixo-bound and Ixo/LY2119620-bound states.

Color code: gray (apo), light teal (ACh), purple (ACh/LY2119620), blue (Ixo), light green (Ixo/LY2119620). **e** Comparison of conformations of intracellular residues $R121^{3.50}$, $Y206^{5.58}$ and $Y440^{7.53}$ in G-protein and $\beta$-arrestin-1 coupled structures. PDB code: Ixo-M2R-$\beta$-arrestin-1 (6U1N), Ixo-M2R-GoA (6OIK). **f** Effects of $\beta$-arrestin-1 and miniG$\alpha$o on the HSQC spectra of $M202^{5.54}$. Color code: blue (Ixo), magenta (Ixo/phosphorylation), red (Ixo/miniG$\alpha$o), green (Ixo/phosphorylation/$\beta$-arrestin-1).

had a smaller inward displacement compared to the G-protein-bound structure (Fig. 7e), suggesting that $\beta$-arrestin and G-protein stabilize different intracellular conformations of M2R.

**Go and $\beta$-arrestin-1stabilize distinct conformations in the M2R**
As the cryo-EM map of M2R-$\beta$-arrestin-1 complex has limited resolution (4 Å) and the conformational differences with the G-protein-bound structure were relatively small (Fig. 7e), we sought to use NMR spectroscopy to further examine whether G-protein and $\beta$-arrestin stabilize distinct conformational states of M2R by monitoring the spectral changes of $M202^{5.54}$. The structure of M2R-$\beta$-arrestin-1 was obtained through the fusion of phosphorylated vasopressin 2 receptor (V2R) C-terminal peptide[17]. However, unlike the V2R, the M2R has a long intracellular loop 3 (ICL3) and a short C-terminus. Functional and biochemical studies have shown that phosphorylation of ICL3 play key roles in the interaction between M2R with arrestins[58,59]. In the following NMR studies, we used the full-length M2R (M2RFL) with mutations at the three native methionines in ICL3, on top of the 5 methionine mutations we previously made in the TM region; the resulting construct is named M2RFL$_\triangle$8 M (Supplementary Fig. 1c). The purified receptor in detergent can be efficiently phosphorylated by GRK2 in a ligand-dependent manner as shown by the Pro-Q staining and ion-exchange chromatography (Supplementary Figs. 1b, 7a). Considering the transient nature of receptor-$\beta$-arrestin complex, we used Ixo for the following NMR studies on signaling complexes because of its supraphysiological effects. In addition, we used miniG$\alpha$o as a substitution of the heterotrimeric G-protein to minimize the effects of line broadening due to increased mass (Supplementary Fig. 7b). We found that phosphorylated Ixo-bound M2RFL$_\triangle$8 M can form a SEC-stable

complex with $\beta$-arrestin-1 containing three alanine mutations (I386A, V387A and F388A) that enhance interactions with GPCRs[60] (Supplementary Fig. 7c).

We collected NMR spectra of Ixo-bound M2RFL$_\triangle$8 M alone and in the presence of excess miniG$\alpha$o or $\beta$-arrestin-1 (Supplementary Fig. 7d–j). The overall spectral changes of M2RFL$_\triangle$8 M in response to Ixo or Ixo plus miniG$\alpha$o are essentially similar to those of the ICL3 truncated receptor[29]. Notably, we observed differences in the spectra of phosphorylated and $\beta$-arrestin-1-bound M2RFL$_\triangle$8 M. $M202^{5.54}$ of phosphorylated M2RFL$_\triangle$8 M exhibited a different chemical shift from that of the unphosphorylated receptor (Fig. 7f), suggesting the appearance of a different local environment surrounding $M202^{5.54}$ upon phosphorylation, even though $M202^{5.54}$ would not be expected to directly interact with phosphorylation sites in ICL3. This observation is similar to the recent NMR studies on the $\beta$2AR, where phosphorylation-induced spectral changes were observed for the homologous residue $M215^{5.54}$ [61]. These structural changes in the TM region may be required for the core engagement of $\beta$-arrestin-1. While the $^{13}CH_3$-$\epsilon$-Met resonances of extracellular probes $M77^{2.58}$ and $M406^{6.54}$ disappeared upon $\beta$-arrestin-1 binding due to the increased mass, the resonance for $M202^{5.54}$ was readily detectable (Supplementary Fig. 7d–j). Upon coupling with $\beta$-arrestin-1, we observed further spectral changes for $M202^{5.54}$ (Fig. 7f). In agreement with the structural differences observed in cryo-EM structures (Fig. 7e), the $\beta$-arrestin-1-bound $M202^{5.54}$ peak positioned between the Ixo-bound and Ixo plus miniG$\alpha$o-bound peaks (Fig. 7f), indicating a distinct and/or more occluded fully active conformation stabilized by $\beta$-arrestin−1 as compared to G-protein. The heterogeneity of the $\beta$-arrestin-1-bound $M202^{5.54}$ peak suggests the existence of slow conformational exchange between two states within the complex or maybe due to the dissociation of some of the complex during data acquisition.

## Discussion

In this study, we determined high-resolution structures of the M2R-GoA signaling complex bound to the endogenous agonist ACh in the presence and absence of the PAM LY2119620. These structural snapshots, together with NMR spectroscopic studies show that ACh is less efficacious than Ixo at stabilizing a uniform nucleotide-free M2R-GoA signaling complex, which likely results from their distinct interactions with the orthosteric pocket (Fig. 3h, i). The diverse coupling modes between the M2R and GoA reveal the dynamic nature of the signaling complex, providing insights into the functional differences between ACh and Ixo, as well as the coupling specificity of mAChRs (Fig. 5). These findings, together with our previous NMR studies on the ligand-dependent conformational changes in the receptor transmembrane core[29], reveal how agonists interacting differentially with the orthosteric pocket can stabilize diverse intracellular G-protein coupling modes and ultimately distinct functional outcomes. It should be noted that current cryo-EM structures only show subtle differences in the G-protein conformation of the M2R-GoA complex with different ligands (Supplementary Fig. 4f). Further biophysical studies are required to determine if different ligand efficacies could be associated with distinct G-protein conformational dynamics (i.e. the movement of α-helical domain).

The present study also shows that the PAM LY2119620 not only enhances agonist binding, but also has positive allosteric effects on β-arrestin recruitment and negative allosteric effects on G-protein activation (Fig. 1). Thus, LY2119620 may modulate the signaling bias of orthosteric agonists, an interesting phenomenon that has also been observed for several other GPCRs[25], yet the underlying mechanism is poorly understood. Both cryo-EM and NMR spectroscopic data show that LY2119620 stabilizes distinct conformational changes in the extracellular domain of agonist-bound or agonist plus GoA-bound M2R, particularly in the aromatic network that links the orthosteric and allosteric sites (Fig. 6e–g), which is likely responsible for the enhanced binding of both ACh and Ixo. While both the current cryo-EM structures and the previous crystal structures with or without LY2119620 show nearly identical intracellular conformations[15], NMR spectroscopy reveals that LY2119620 bound together with an agonist can stabilize distinct, possibly more occluded active conformations in the intracellular domain compared to those bound to agonist alone (Fig. 7c, d). These differences could explain the different allosteric effects of LY2119620 on β-arrestin recruitment and G-protein activation. Taken together, our study provides valuable structural and dynamic information for understanding the complex signaling behavior of the M2R. A better understanding of different ligand efficacies, allosteric regulation, and signaling bias may facilitate the development of safer and more effective therapeutics targeting mAChRs.

## Methods

### Protein expression and purification for cryo-EM

The construct of M2R for cryo-EM experiments in this study is the same as previously described for crystallization studies, where the intracellular loop 3 was truncated (Supplementary Fig. 1c)[15]. The modified M2R gene was cloned to pFastBac vector with an N-terminal FLAG tag and a C-terminal 8x histidine tag to express in Sf9 insect cells. Sf9 cells were grown in SIM SF Medium (Sino Biological Inc.) and were infected with recombinant baculovirus at a density of $4 \times 10^6$ cells mL$^{-1}$ in the presence of 5 µM atropine. After 48 h infection at 27 °C, the cells were spun down and cell pellets were stored at −80 °C until use. Thawed cell pellets were resuspended in lysis buffer (10 mM Tris, 1 mM EDTA, 100 µM ACh, 2.5 µg mL$^{-1}$ leupeptin, 160 µg mL$^{-1}$ benzamidine). Cell membranes were spun down and solubilized with 20 mM HEPES pH 7.5, 750 mM NaCl, 1% DDM, 0.2% sodium cholate, 0.03% CHS, 30% glycerol, 100 µM ACh, 2.5 µg mL$^{-1}$ leupeptin, and 160 µg mL$^{-1}$ benzamidine. Nickel-NTA sepharose was added to the solubilized receptor and incubated for 2 h at 4 °C. The resin was spun down and washed in

batch for three times with 20 mM HEPES pH 7.5, 750 mM NaCl, 0.1% DDM, 0.02% sodium cholate, 0.03% CHS, 30% glycerol, 100 µM ACh, 2.5 µg mL$^{-1}$ leupeptin and 160 µg mL$^{-1}$ benzamidine. The washed resin was poured into a glass column and the receptor was eluted in the wash buffer supplemented with 250 mM imidazole. The Ni-NTA chromatography purified receptor was then loaded onto a column with anti-flag M1 affinity resin and gradually exchanged into a buffer containing 20 mM HEPES pH 7.5, 100 mM NaCl, 0.01% lauryl maltose neopentyl glycol (MNG), 0.003% CHS, 100 µM ACh supplemented with 2 mM CaCl$_2$, and then eluted with same buffer containing 0.2 mg mL$^{-1}$ flag peptide and 5 mM EDTA. The flag affinity chromatography purified receptor was then purified by size exclusion chromatography (SEC) against 20 mM HEPES pH 7.5, 100 mM NaCl, 0.01% MNG, 0.003% CHS, and 100 µM ACh. The monodisperse peak fractions were collected and concentrated using a 50 kilo Dalton (kDa) molecular weight cutoff Millipore concentrator. The concentrated receptor was stored at −80 °C for complex formation.

The heterotrimeric GoA protein was expressed and purified similarly as previously described[29]. In brief, the protein was purified with Ni-NTA and MonoQ chromatography. The peak fractions from MonoQ were collected and exchanged to 20 mM HEPES pH 7.5, 100 mM NaCl, 0.1% DDM, 1 mM MgCl$_2$, 10 µM GDP, 50 µM TCEP by repeated concentration and dilution using a 50 kDa molecular weight cutoff Millipore concentrator. The concentrated heterotrimeric GoA was aliquoted, flash frozen in liquid nitrogen and stored at −80 °C for complex formation.

The scFv16 protein was purified as described[62]. Briefly, the scFv16 gene was cloned into pFastBac vector with a C-terminal histidine tag and expressed in secreted form in HighFive insect cells using bac-to-bac system. The protein was purified with Ni-NTA chromatography and SEC in 20 mM HEPES, pH 7.5, 100 mM NaCl. The monomeric peak fractions were pooled, concentrated and stored at −80 °C for complex formation.

The ACh-M2R-GoA-scFv16 complex was formed in a same manner as described for Ixo-M2R-GoA complex[16]. Briefly, purified receptor and GoA was mixed at 1:1.2 molar ratio in 20 mM HEPES pH 7.5, 100 mM NaCl, 1% MNG, 1 mM MgCl$_2$, 10 µM GDP, and 1 mM ACh. After 3 h incubation on ice, the complex was then treated with 50 units of apyrase (NEB) on ice overnight, and then purified with an anti-Flag M1 column into a buffer containing 20 mM HEPES, pH 7.5, 100 mM NaCl, 0.0075% MNG, 0.0025% GDN, and 0.001% CHS, 1 mM ACh and 2 mM CaCl$_2$. After elution by adding 5 mM EDTA and 0.2 mg mL$^{-1}$ Flag peptide, the complex was concentrated and incubated with excess scFv16 for 1 h on ice, then loaded onto Superdex 200 Increase 10/300GL column against 20 mM HEPES, pH 7.5, 100 mM NaCl, 0.00075% MNG, 0.00025% GDN, 0.0001% CHS, and 1 mM ACh. The monomeric peak fraction of the ACh-M2R-GoA-scFv16 complex was collected and concentrated to 2–4 mg mL$^{-1}$ using a 100 kDa molecular weight cutoff Millipore concentrator for cryo-EM experiments.

### Cryo-EM sample preparation and data collection

3 µL complex sample was dropped onto the grid glow discharged using easiGlow™ Glow Discharge Cleaning System (PELCO, USA) and then blotted for 3.5 s with blotting force 0, and plunged into liquid ethane cooled by liquid nitrogen using Vitrobot Mark IV (Thermo Fisher Scientific, USA). For LY2119620-bound sample, the complex was incubated with 250 µM LY2119620 at RT for 20 min first and then frozen in the same manner. The movies were collected with the 300 kV Titan Krios Gi3 microscope by Gatan K3 BioQuantum Camera at the magnification of 105,000, with a pixel size of 0.85 Å. Inelastically scattered electrons were excluded by a GIF Quantum energy filter (Gatan, USA) using a slit width of 20 eV. The movies were acquired with the defocus range of −1 to −2 µm with total exposure time of 2.5 s fragmented into 50 frames and with the dose rate of 17.3 e/pixel/s using SerialEM3.7.

## Image processing and model building

For ACh-M2R-GoA-scFv16 complex, a total of 1761 image stacks were collected and subjected for motion correction using MotionCor2[63]. Contrast transfer function parameters were estimated by CTFFIND4[64], implemented in RELION[65]. 900741 particles were auto-picked from micrographs by RELION and then subjected to 2D classification in cryoSPARC[66]. Selected particles with an appropriate 2D average were further subjected to 3D classification in RELION. A dataset of 264600 particles for S1 state and a dataset of 93148 particles for S2 state were selected and then subjected to cryoSPARC for 3D reconstruction. Subsequent non-uniform refinement and local refinement yielded final maps with global resolution of 3.21 Å for S1 state and 3.3 Å for S2 state at FSC 0.143. Local resolution maps were generated in cryoSPAR using Local Resolution Estimation.

For LY2119620-bound complex, a total 3285 image stacks were collected and processed using same strategy as described above. Finally, a dataset of 92571 particles for S1 state and a dataset of 80017 particles for S2 state were used for 3D reconstruction in cryoSPARC. Non-uniform refinement and local refinement yielded final maps with global resolution of 3.16 Å for S1 state and 3.22 Å for S2 state at FSC 0.143. Local resolution maps were generated in cryoSPAR using Local Resolution Estimation. Automatic local sharpening in DeepEMhancer[67] were performed for both EM density maps to optimize local density.

The coordinate of Ixo-M2R-GoA (PDB 6OIK])[16] was used as initial template by removing ligands from the model. The template was docked into each EM density map using Chimera[68], followed by iterative manual refinement in Coot[69] and real space refinement in Phenix[70]. The final model statistics was validated by Molprobity[71].

## Expression and purification of miniGαo

The miniGαo sequence was designed as described[72] and cloned into pET21a vector containing an N-terminal, 3 C protease-cleavable 8xHis-tag. Plasmids were transformed into *E. coli* BL21(DE3) cells. Cells were grown to A600 = 0.8 at 37 °C in TB media containing 100 µg mL$^{-1}$ ampicillin. Cells were then induced by addition of 0.2 mM IPTG and were incubated overnight at 22-23 °C. Cells were harvested by centrifugation and stored at −80 °C until use. Cell pellets were resuspended in buffer A composed of 20 mM HEPEs pH 7.5, 100 mM NaCl, 10% glycerol, 10 mM imidazole, 5 mM MgCl2, 5 mM 2-Mercaptoethanol (β-ME), 10 µM GDP, 2.5 µg mL$^{-1}$ leupeptin and 160 µg mL$^{-1}$ benzamidine. Cells were then broken by sonication for 5 min on the ice-water bath. Cell debris was removed by centrifugation at 12,000×g and kept the supernatant. Ni-NTA resin pre-equilibrated in buffer A were added to the supernatant and shake for 1.5 h at 4 °C. After incubation, the Ni-NTA resin was spun down and poured into a glass column, and then washed with 50-100 ml buffer A containing 20 mM imidazole and 50-100 mL buffer B composed of 20 mM HEPEs pH 7.5, 500 mM NaCl, 10% glycerol, 10 mM imidazole, 5 mM MgCl2, 5 mM β-ME, and 10 µM GDP. The miniGoa was then eluted with buffer C composed of 20 mM HEPEs pH 7.5, 100 mM NaCl, 10% glycerol, 250 mM imidazole, 5 mM MgCl2, 5 mM β-ME, and 10 µM GDP. The 8xHis-tag was removed using 3 C protease. Cleaved miniGαo was purified by an additional negative Ni-NTA purification step. The Ni-NTA chromatography purified protein was further purified with SEC with buffer D composed of 20 mM HEPEs pH 7.5, 100 mM NaCl, 5 mM MgCl2, 50 µM TCEP, and 10 µM GDP. For NMR experiments, the buffer D was prepared in D2O (>99%). The monodisperse peak fractions of miniGαo was collected and then concentrated using a 10 kDa molecular weight cutoff Millipore concentrator. The concentrated miniGαo was aliquoted, flash frozen in liquid nitrogen and stored at −80 °C before use.

## Expression and purification of GRK2

Bovine GRK2 with 6xHis-tag was cloned into pFastBac and expressed in SF9 insect cells. SF9 cells were grown to a density of $4 \times 10^6$ cells mL$^{-1}$ and then infected with GRK2 baculovirus at a ratio of 10−20 ml L$^{-1}$. After 48 h incubation at 27 °C, the infected cells were harvested by centrifugation and stored at −80 °C until use. Cell pellets were resuspended lysis buffer composed of 20 mM HEPES, 250 mM NaCl, 1 mM β-ME, 0.02% Triton-X100, 2.5 µg mL$^{-1}$ leupeptin and 160 µg mL$^{-1}$ benzamidine and were stirred at RT for 15 min. Cells were then broken by sonication for 5 min on ice-water bath. Cell debris was removed by centrifugation at 12,000×g and the supernatant was collected. Ni-NTA resin pre-equilibrated in lysis buffer were added to the supernatant and shake for 1.5 h at 4 °C. After incubation, the Ni-NTA resin was spun down and poured into a glass column, and then washed with 50 mL lysis buffer and then eluted with lysis buffer containing 250 mM imidazole. The Ni-NTA purified GRK2 was finally purified by SEC against 20 mM HEPES pH 7.5, 100 mM NaCl. The monodisperse peak fractions was collected and then concentrated using a 50 kDa molecular weight cutoff Millipore concentrator. The concentrated GRK2 was aliquoted, flash frozen in liquid nitrogen and frozen at −80 °C until use.

## Expression and purification of β-arrestin-1

The full-length β-arrestin-1 with 3 alanine mutations (I386A, V387A and F388A) was cloned into pET21a vector containing an N-terminal, 3 C protease-cleavable 6xHis-tag. Plasmids were then transformed into *E. coli* BL21(DE3) cells. Cells were grown to A600 = 2.0 at 37 °C in TB media containing 50 µg mL$^{-1}$ ampicillin. Cells were then induced by addition of 50 µM IPTG and were incubated overnight at 22–23 °C. Cells were harvested by centrifugation and stored at −80 °C until use.

Cell pellets were resuspended in buffer A composed of 20 mM HEPEs pH 7.5, 500 mM NaCl, 10% glycerol, 10 mM imidazole, 5 mM β-ME, 2.5 µg mL$^{-1}$ leupeptin and 160 µg mL$^{-1}$ benzamidine. Cells were then broken by sonication for 5 min on ice-water bath. Insoluble factions were removed by centrifugation at 12000 x g and the supernatant was collected. Ni-NTA resin pre-equilibrated in buffer A were added to the supernatant and shake for 1.5 h at 4 °C. After incubation, the Ni-NTA resin was spun down and poured into a glass column, and then washed with 50–100 mL buffer A containing 20 mM, 30 mM, 40 mM imidazole sequentially. The β-arrestin-1 was then eluted with buffer A containing 250 mM imidazole. The 6xHis-tag was removed using 3 C protease. Cleaved β-arrestin-1 was purified by an additional negative Ni-NTA purification step. The Ni-NTA purified β-arrestin-1 was further purified with SEC against 20 mM HEPES pH 7.5, 100 mM NaCl. For NMR experiments, the buffer B was prepared in D2O (>99 %). The monodisperse peak fractions of β-arrestin-1 was collected and then concentrated using a 30 kDa molecular weight cutoff Millipore concentrator. The concentrated β-arrestin-1 was aliquoted, flash frozen in liquid nitrogen and stored at −80 °C before use.

## Phosphorylation of full-length M2R

We used a full-length construct M2RFL△8 M for complexing with miniGαo and β-arrestin-1. The receptor was purified as described above for cryo-EM sample. Excess GRK2 was mixed with the M2R in a buffer composed of 20 mM HEPES pH 7.5, 100 mM NaCl, 0.01% MNG, 0.003% CHS, 500 µM ATP in the presence and absence of 100 µM Ixo or 1 mM carbachol to initiate the phosphorylation at room temperature. The extent of phosphorylation at various time points (from 0 to 120 min) was evaluated by Pro-Q and Coomassie blue staining. The final phosphorylated receptor (Ixo-bound) was further evaluated by ion-exchange chromatography. The complex formation test was performed as follows: unphosphorylated Ixo-bound M2RFL△8 M or phosphorylated M2RFL△8 M was mixed with miniGαo or β-arrestin-1, respectively and incubated on ice for 30 min. The receptor alone or mixed complex sample were then analyzed by SEC against 20 mM HEPES pH 7.5, 100 mM NaCl, 0.01% MNG, 0.003% CHS in the presence of 10 µM Ixo.

## NMR spectroscopy

Three different constructs were used for NMR experiments in this study. The M2Rmini$_\triangle$5M construct was the same as previously described[29] and used for testing the effect of the PAM LY2119620. The M2RFL$_\triangle$8M construct was used to test the effect of phosphorylation and downstream signaling proteins. The Met383 probe on TM6 was introduced based on the M2RFL$_\triangle$8M construct, in which the following 9 methionine mutations were introduced to minimize the spectral overlapping: M01T, M45L, M139L, M142L, M456T, M248L, M296L, M368L and M143L[29]. The construct for Met383 probe was thereby named as M2RFL$_\triangle$9M_Met383.

The $^{13}CH_3$-ε-methionine labeled M2R was expressed, labeled and purified as previously described[29]. In Brief, sf9 cells were grown in methionine-deficient medium and infected at a density of $4 \times 10^6$ ml$^{-1}$ with M2R baculovirus in the presence of 10 μM atropine. After adding $^{13}CH_3$-ε-methionine at a concentration of 250 mg L$^{-1}$, the sf9 cells were incubated for 48 h at 27 °C and then harvested. The receptor was purified by Ni-NTA chromatography, flag affinity chromatography and SEC and exchanged to a $D_2O$-based buffer containing 20 mM HEPES pH 7.5, 100 mM NaCl, 0.01% LMNG, 0.003% CHS. The receptor was finally concentrated to around 80 μM for NMR experiments. The heterotrimeric GoA protein for NMR experiments was also purified in a $D_2O$-based buffer containing 20 mM HEPES pH 7.5, 100 mM NaCl, 0.02% LMNG, 1 mM $MgCl_2$, 10 μM GDP, 50 μM TCEP.

The apo-state, ACh-bound, and Ixo-bound spectra of M2Rmini$_\triangle$5M were repeated by following previously described methods. Both agonists were added to the NMR sample at concentrations of at least 10-fold stoichiometric excess over the receptor (around 1 mM), and more than 10-fold over their Ki values to ensure that the receptors were fully occupied by the ligands. For LY2119620-bound spectra, LY2119620 was added into the apo-state receptor at a concentration of 1 mM. For LY2119620 and (ACh or Ixo)-bound spectra, LY2119620 was added into the agonist-bound sample, incubated for 30 min at room temperature and then subjected to NMR experiments. The spectra of M2RFL$_\triangle$8M and M2RFL$_\triangle$9M_Met383 in different conditions were collected in a same way as those of M2Rmini$_\triangle$5M. For complex samples, the miniGαo, β-arrestin-1 or heterotrimeric GoA was added to the agonist-bound receptor at a molar ratio of 1:1.5 for data collection. All NMR samples were loaded into the Shigemi microtubes for data collection at 25 °C on a Bruker Avance 800-MHz spectrometer equipped with a triple-resonance cryogenic probe. The $^1H$-$^{13}C$ hetero-nuclear single-quantum coherence (HSQC) spectra were recorded with spectral widths of 12820.5 Hz in the 1H-dimension (ω1) and 16077.2 Hz in the 13C-dimension (ω1) centered at 45 ppm in $^{13}C$-dimension. For all spectra, 512 × 128 complex points were recorded and a relaxation delay of 2 s were inserted to allow spin to relax back to equilibrium. For M2Rmini$_\triangle$5M, 80 scans gave rise to an acquisition time around 12 h for each spectrum. For M2RFL$_\triangle$8M and M2RFL$_\triangle$9M_Met383, 160 scans gave rise to an acquisition time -24 h for each spectrum. All NMR spectra were processed using the software package NMRPipe/NMRDraw and analyzed using the program NMRViewJ. All spectra were normalized using a natural abundance peak from a highly flexible methyl group in M2R at around 1.20 ppm in the $^1H$ dimension and 18.00 ppm in the $^{13}C$ dimension as previously described[29].

## G-protein IP-1 assay and β-arrestin recruitment assay

Measurement of G-protein mediated activation of M2R was performed applying the IP-One HTRF® accumulation assay (Cisbio, Codolet, France) as previously described protocols[29,73]. In brief, HEK 293T cells were co-transfected with the cDNA coding for M2R and the hybrid G-protein Gαqi (Gαq protein with the last five amino acids at the C-terminus replaced by the corresponding sequence of Gαi; gift from The J. David Gladstone Institutes, San Francisco, CA) and transferred into 384-well micro plates. On the day of the experiment, cells were pre-incubated with the allosteric modulator (fixed concentration in the

range of 0.1 to 100 μM) for 30 min before adding the agonist (final concentration for Ixo: 0.01 pM to 1 μM, for Ach: 10 pM to 300 μM) and incubating for 90 min. Accumulation of second messenger was stopped by addition of the detection reagents (IP1-d2 conjugate and Anti-IP1cryptate TB conjugate). After an additional 60 min, time-resolved fluorescence resonance energy transfer was measured at 620 nm and 665 nm using the Clariostar plate reader (BMG, Ortenberg, Germany). The obtained FRET-ratio was normalized to the maximum effect of reference ACh (100%) and vehicle (0%). Normalized concentration-response curves from four to eight experiments each done in duplicates were analyzed using the algorithms for four parameter non-linear regression implemented in PRISM 8.0 (GraphPad Software, USA) to derive $EC_{50}$ and $E_{max}$ values.

Determination of β-arrestin-2 recruitment was performed applying the PathHunter assay (DiscoverX, Birmingham, U.K.) which is based on fragment complementation of β-galactosidase as described. In detail, HEK293T cells were transfected with the flag-tagged M2 receptor carrying the PKA fragment for enzyme complementation and transferred into 384-well micro plates. At the day of measurement allosteric modulator at a distinct concentration was added to the cells and preincubated for 30 min followed by addition of the agonist (for Ixo: 0.1 pM to 1 μM; for ACh: 1 nM to 300 μM) and incubation for further 90 min. Determination of chemoluminescence was done with a Clariostar plate reader (BMG, Ortenberg, Germany). Data analysis was performed by normalizing the raw data relative to basal activity (0%) and the maximum effect of the reference agonist ACh (100%). Normalized curves from three to five individual experiments each done in duplicate were analyzed by non-linear regression applying the algorithms in Prism 8.0 (GraphPad, San Diego, CA) to get dose-response curves representing average $EC_{50}$ and Emax value.

Significance of Emax for the agonists Ach or Ixo alone with agonist in the presence of a distinct concentration of LY2119620 in both signaling pathways was analyzed by non-paired comparison by One-way ANOVA applying Dunnett's multiple comparisons test in Prism 8. The threshold for significance was set as 95% confidence interval and displayed as $p$-value ($p < 0.05$).

## GTPase Glo™ assay

The receptors for GTPase-Glo™ assay were expressed and purified as described above and frozen at −80 °C before use. For time course experiments, the GTPase reaction was initiated by mixing GoA and M2R in 5 μL reaction buffer (20 mM HEPES, 100 mM NaCl, 0.01% MNG, 1 mM $MgCl_2$, 5 μM GTP, 5 μM GDP, 100 μM ACh or 10 μM Ixo) in a 384-well plate. GoA and M2R were fixed at a final concentration of 0.3 μM and 1 μM respectively. For every independent experiment, GoA alone was set as a reference. The GTPase reaction was incubated at room temperature (22–25 °C) for different times (15–90 min). For single points experiments with LY2119620, GoA and M2R were fixed at a final concentration of 0.5 μM and 1 μM respectively and the GTPase reaction was initiated in a same way by adding 100 μM LY2119620 to the agonist-bound sample and incubated at room temperature for 90 min. After incubation, 5 μL reconstituted 1xGTPase-Glo™ Reagent (Promega) was added to the completed GTPase reaction, mixed briefly and incubated with shaking for 30 min at room temperature (22–25 °C) to convert the remaining GTP into ATP. Then 10 μL Detection Reagent (Promega) was added to the system and incubated for 5-10 min at room temperature (22–25 °C) to convert the ATP into luminescent signals. Luminescence intensity was quantified using a Multimode Plate Reader (PerkinElmer) luminescence counter. Data were normalized to GoA reference and analyzed using GraphPad Prism 9.2.0.

## Radio-ligand binding assay

Purified M2R receptor was reconstituted into high-density lipoprotein (HDL) particles constituting apolipoprotein A1(APOA1) and mixture of POPG: POPC lipids with 3:2 (mol:mol)ratio as previously described[74].

For competition binding studies, Sf9 membranes or M2R receptor reconstituted in HDL particle containing 50-100 femtomoles were incubated with 2 nM [³H]-NMS, in the presence or absence of heterotrimeric GoA, and increasing concentration of test ligand (ACh or Ixo) in a buffer containing 20 mM HEPES pH 7.5, 100 mM NaCl and 0.5% bovine serum albumin. GoA titration competition binding assay was carried out by incubating the Sf9 membranes with a fixed concentration of [³H]-NMS and agonist with varying concentration of purified GoA for 2 h. Membranes or HDL particles were separated from excess [³H]-NMS by Whatman GF/B filters using a Brandel 48-well harvester. The bound radioligand were read on a liquid scintillation counter (MicroBeta Jet, PerkinElmer). Data were analyzed by GraphPad Prism 9.2.0.

### NanoBiT-G-protein dissociation assay

M2R ligand-induced GoA activation was measured by the NanoBiT-G-protein dissociation assay (Inoue et al., 2019) with minor modifications. Plasmid transfection for HEK293A cells (Thermo Fisher Scientific) was performed by combining 25 μl (volume is per 10-cm culture dish) of polyethylenimine solution (1 mg/mL) and a mixture of plasmids each encoding the large fragment (LgBiT)-containing the GαoA subunit (500 ng), the small fragment (SmBiT)-fused Gγ2 subunit with the C68S mutation (2.5 μg) and the untagged Gβ1 subunit (2.5 μg), along with a human, wild-type M2R (1 μg; containing N-terminal HA-derived signal sequence followed by the FLAG-epitope tag). After an incubation for one day, the transfected cells were harvested, pelleted with centrifugation, and suspended in 9 ml of Hank's balanced saline solution (HBSS) containing 0.01% bovine serum albumin (BSA fatty acid–free grade, SERVA) and 5 mM HEPES (pH 7.4) (assay buffer). The cell suspension was dispensed in a white 96-well plate at a volume of 70 μl per well and mixed with 20 μl of 50 μM coelenterazine (Carbosynth) diluted in the assay buffer. After 2 h incubation at room temperature, the plate was measured for baseline luminescence (SpectraMax L, Molecular Devices) and 10 μl of titrated concentrations of LY2119620 diluted in the assay buffer were added. After 10 min, 20 μl of titrated concentrations of ACh or Ixo diluted in the assay were manually added and the plate was positioned for luminescent measurement. Kinetics data points from 5 min to 10 min were averaged and normalized to the initial count and used as a G-protein dissociation index. For the mutant study, the cell culture and the transfection mixture were scaled down to a 6-cm dish (4 ml culture media) and the transfected cells were suspended in 4 ml of the assay buffer and seeded in the 96-well plate at a volume of 80 μL. The G-protein dissociation signals were fitted to a four-parameter sigmoidal concentration-response curve (GraphPad Prism8). For each experiment, we calculated $E_{max}$ by normalizing Span to the LY2119620-free condition and $\Delta pEC_{50}$ by subtracting $pEC_{50}$ of the LY2119620-free condition performed in parallel and used the values to denote allosteric modulator activity. Similarly, for the mutant study, we calculated $\Delta pEC_{50}$ by subtracting $pEC_{50}$ of the WT (1:1) condition performed in parallel.

### Flow cytometry

Transfection was performed according to the same procedure as described in the "NanoBiT-G-protein dissociation assay" section, except that the cell culture and the transfection mixture were scaled down to a 6-well plate (2 ml culture media). One day after transfection, the cells were collected by adding 200 μl of 0.53 mM EDTA-containing Dulbecco's PBS (D-PBS), followed by 200 μl of 5 mM HEPES (pH 7.4)-containing Hank's Balanced Salt Solution (HBSS). The cell suspension was transferred to a 96-well V-bottom plate in duplicate and fluorescently labeled with an anti-FLAG epitope (DYKDDDDK) tag monoclonal antibody (Clone 1E6, FujiFilm Wako Pure Chemicals; 10 μg per ml diluted in 2% goat serum- and 2 mM EDTA-containing D-PBS (blocking buffer)) and a goat anti-mouse IgG secondary antibody conjugated with Alexa Fluor 488 (Thermo Fisher Scientific, 10 μg per ml diluted in the blocking buffer). After washing with D-PBS, the cells were

resuspended in 200 μl of 2 mM EDTA-containing-D-PBS and filtered through a 40-μm filter. The fluorescent intensity of single cells was quantified by an EC800 flow cytometer equipped with a 488 nm laser (Sony). The fluorescent signal derived from Alexa Fluor 488 was recorded in an FL1 channel, and the flow cytometry data were analyzed with the FlowJo software (FlowJo). Live cells were gated with a forward scatter (FS-Peak-Lin) cutoff at the 390 setting, with a gain value of 1.7. Values of mean fluorescence intensity (MFI) from ~20,000 cells per sample were used for analysis. Typically, we obtained a WT MFI value of 2000 (arbitrary unit) and a mock MFI value of 20. For each experiment, we normalized an MFI value of the mutants by that of the WT (1:1) condition performed in parallel and denoted relative levels.

### Reporting summary

Further information on research design is available in the Nature Portfolio Reporting Summary linked to this article.

## Data availability

The structural data generated in this study have been deposited in the Protein Data Bank (PDB) under accession codes 7T8X, 7T90, 7T94, and 7T96. The 3D cryo-EM maps have been deposited in the Electron Microscopy Data Bank (EMBD) under accession codes EMDB-25748, EMDB-25749, EMDB-25751, and EMDB-25752. The functional and pharmacological data generated in this study are compiled in the Source Data file provided with this paper. Source data are provided with this paper.

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

## Acknowledgements

We thank the Cryo-Electron Microscopy Center, the Chinese University of Hong Kong (Shenzhen) for their support of cryo-EM data collection; Kayo Sato, Shigeko Nakano, and Ayumi Inoue at Tohoku University for the NanoBiT assay and the flow cytometry experiment. This work was supported by the Beijing Advanced Innovation Center for Structural Biology, School of Medicine, Tsinghua University; by the German Research Foundation (GRK1910 to P.G.) and by the American Heart Association Postdoctoral Fellowship (H.W.). All NMR experiments were performed at the Beijing NMR Center and the NMR facility of the National Center for Protein Sciences at Peking University, supported by the Grant 2016YFA0501201 from the National Key R&D Program of China to C. J.; Y.D. is supported by a grant from Science, Technology and Innovation Commission of Shenzhen Municipality (JCYJ20200109150019113), Shenzhen Bay Open Project (SZBL2020090501011) and a Presidential Fellowship at the Chinese University of Hong Kong (Shenzhen). A.I. was funded by the PRIME 19gm5910013, the LEAP 20gm0010004, and the BINDS JP20am0101095 from the Japan Agency for Medical Research and Development (AMED); KAKENHI 21H04791, 21H051130, JPJSBP120213501, and JPJSBP120218801 from by the Japan Society for the Promotion of Science (JSPS); FOREST Program JPMJFR215T and JST Moonshot Research and Development Program JPMJMS2023 from Japan Science and Technology Agency (JST); Daiichi Sankyo Foundation of Life Science; Takeda Science Foundation; Ono Medical Research Foundation; The Uehara Memorial Foundation. Brian Kobilka is a Chan Zuckerberg Biohub Investigator.

## Author contributions

J.X. prepared all NMR samples, processed and analyzed NMR data, froze the grids, collected and processed cryo-EM data, built the models with inputs from H. W. and S. M., analyzed the structures. Q. W. expressed and purified the M2R receptor and G-protein, assembled the M2R-GoA complex and participated grids freezing and cryo-EM data collection. H.H., J.X., and A.I. performed functional assays. Y.H and X.N. conducted the NMR experiments and assisted with NMR data processing and analysis. P.G. supervised G-protein IP-1 and β-arrestin recruitment assays. Y.D. and Y. T. supervised cryo-EM sample preparation. S. M. expressed and purified the miniGαo protein. C.J. supervised NMR experiments and data processing. B.K.K. provided overall project supervision. J.X. and B.K.K. wrote the manuscript with contributions from all authors.

## Competing interests

Brian Kobilka is a co-founder and consultant for ConfometRx. The remaining authors declare no competing interests.
