## [Peer Review File · Nature Communications]

Structural and dynamic insights into supra-physiological activation and allosteric modulation of a muscarinic acetylcholine receptorREVIEWER COMMENTS

Reviewer #1 (Remarks to the Author):

This is an outstanding and comprehensive study, which delves into the structure-function relationship regarding the endogenous agonist of the GPCR, M2R, and a supramolecular agonist. Importantly, comparative cryo-EM studies were performed in complex with G protein both with GDP and after removal of nucleotide. The functional studies and NMR studies, the G protein activation assays and arrestin-1 assays all come together beautifully to connect structure, conformational heterogeneity, and function. Notably, with endogenous agonist the authors observe two very distinct activation states via cryo-EM, which they argue are both fully active signatures. This is not the case with the synthetic super-agonist and there are corresponding observations of the toggle switch and allosteric response at the alpha-5 and alpha-N helices, alongside corroborating ¹³Cmethyl TROSY signatures at 6.31 confirming what cryo-EM identifies. The authors then go on to identify mechanisms of a PAM which biases arresting association. This is important because prior X-ray structures revealed little conformational changes. Both NMR and cryo-EM identify signatures that could explain how this PAM achieves arrestin bias, although this data is slightly more speculative.

The paper is appropriately long and detailed and the comparisons to other muscarinic and NTS1 structures are appropriate and interesting. I think this paper should be published with only very minor corrections. Some grammar corrections are summarized in my attached pdf. (Anything highlighted in pink is awkward grammatically).

Reviewer #2 (Remarks to the Author):

Xu et al. determined the cryo-EM structures of M2R-GoA complexes in the presence of the endogenous agonist acetylcholine or a supra-physiological agonist iperoxo with or without an allosteric modulator LY2119620 bound. These structures reveal that iperoxo stabilizes a uniform M2R-GoA complex, while acetylcholine stabilizes a more heterogeneous complex. Based on the biochemical and signaling assays, LY2119620 seems to be an arrestin-biased allosteric modulator that enhances arrestin recruitment while impairing G protein activation. Structural and NMR spectroscopic data provide interesting insights into mechanisms underlying super-agonism of iperoxo and biased signaling of LY2119620. The manuscript is well written and illustrated. However, I have a few comments that need to be addressed before I can recommend it for publication in Nature Communications:

Major points:

1. For monitoring Gi/o activation by M2R, the authors performed G-protein IP-one accumulation assay with a chimera Gαqi where the last five amino acids at the C-terminus of Gαq are replaced by the equivalent residues of Gαi1. However, the structures of M2R-GoA complex show that in addition to the C-terminal helix (α5) of GαoA, the other region of GαoA is involved in binding M2R. In addition, the last five residues of Gαo are not identical to those of Gαi. I am a bit worried whether Gαqi can recapitulate Gαi/o in vivo?

2. Interpretation of bias signaling should take account into both the potency (EC50) and efficacy (Emax). LY2119620 increases the potencies (EC50) of both Ach and Ixo in the G-protein IP-One assay to a greater extent than those in the arrestin-recruitment assay. Besides, only high concentrations of LY2119620 (30 μM, 100 μM) can decrease the efficacies of Ach and Ixo in the IP-One assay, whereas relatively low concentrations of LY2119620 (0.1-10 μM) show minimal effects on their efficacies while significantly increasing their potencies. These signaling assay results suggest that LY2119620 enhances both the G protein activation and arrestin recruitment. Thus, the claim that LY2119620 impairs G protein activation may be not accurate. I suggest the authors tone down the arrestin-biased activity of LY2119620.

3. It is interesting to obtain two different conformational states of the Ach-bound M2R-GoA complex with two distinct G-protein orientations, which were also observed in the NTSR1-Gi complex. Both M2R and NTSR1 contain a large hydrophobic residue at the position 34.51 of ICL2 (L in M2R, F in NTSR1). The hydrophobic interactions between the residue 34.51 and Gα are stronger in one state than the other state. In fact, most structures of GPCR-Gi/o complex lack these strong hydrophobic interactions. So whether strong hydrophobic interactions between the residue 34.51 and Gαi/o observed in one state (S1 state in M2R) is essential for Gi/o coupling of M2R. Mutagenesis studies are highly encouraged to demonstrate the functional relevance of the two states.

4. Related to Fig. 5e-5g, the concentrations of Ach and Ixo should be provided in the method section or Figure legends. If sub-stoichiometric amount of ligands are added, the two NMR peaks of M383 in M2R when bound to Ach or Ixo should represent the ligand-bound state and apo-state which exchange slowly. If the receptor is saturated by the ligand, the two peaks may represent two different conformational states of M2R when bound to the ligand. The description in the fifth line of PAGE 12 should be revised accordingly.

Minor points:

1. Every data point needs to be provided in graphs, Fig 1m, Fig 1i-l.
2. The 9th line in PAGE 8, W4227.53 should be W4227.35.

3. BW numbers need to be added in Fig. 4.
4. PAGE 17, miniGo α should be miniGo α o.
5. Fig. 2b, the map of Gy is not shown.
6. The outlier residues in Ramachandran plot need to be fixed.
7. EC50 should be changed to EC50 for consistency.

Reviewer #3 (Remarks to the Author):

The manuscript by Xu J. et al. describes structural and dynamic insights into the activation and biased allosteric modulation of the M2 muscarinic acetylcholine receptor. The combination of cryo-electron microscopy (cryoEM) and NMR spectroscopy allows to better understand regulation of this prototypical G protein-coupled receptor (GPCR) by acetylcholine and iperoxo agonists in the presence of the arrestin-biased allosteric modulator LY2119620.

This work is an outstanding study. While both the cryoEM structures and the previous crystal structures with or without LY2119620 show nearly identical intracellular conformations, the NMR spectroscopy reveals in details that the allosteric modulator bound together with an agonist can stabilize distinct active conformations in the intracellular domain compared to those bound to agonist alone. It shows how different structural approaches can be complementary to each other allowing to determine structural snapshots of agonist (+ allosteric modulator)-GPCR-G protein complexes with high resolution together with dynamic information (thanks to the labeling of methionine residues at crucial positions) in different receptor domains. In the future, this structural information will certainly help in the rational design and development of novel therapeutic ligands targeting muscarinic acetylcholine receptors.

The expertise of the authors in the scientific field of GPCRs and structural biology is internationally well renowned, the approaches developed are state-of-the-art and the results presented in the article are convincing and very nicely illustrated. In particular, the sensor positions (labeled methionine) in the M2R are perfectly chosen and the different NMR HSQC spectra are quite clear and unambiguous.

I have no major criticisms; I only have some minor comments.

a) In the Methods section, more precisely in the radioligand binding assay paragraph (page 37), would it be possible to explain how the HDL-reconstituted M2 receptor is prepared? At least, please give a reference or briefly detail the protocol. Only the radioligand binding assay is described, not the purification of the HDL-inserted M2R particle.

b) In the extended data figure 3 panel e-h, it would be important to have an idea of the LY2119620 orientation in the extracellular vestibule. In other words, what are the ECLs seen in these close-up views (probably ECL2 and 3)? Please indicate ECL2 and ECL3, like in the panel a of figure 6.

c) The NMR resonance of K383M is assigned at 2.3 ppm and 17.2 ppm in the ¹³C and ¹H dimension respectively (main text page 12 and extended data figure 4e). This is a mistake. Indeed, it is rather 2.16 ppm and 17.2 ppm in the ¹H and the ¹³C dimension, respectively. Please correct.

d) Is there a contradiction between figure 6 panel d and extended data figure 5 panel b? If well understood, a density for a water molecule is probably seen in the cryoEM-determined S2 conformation of Ach-bound M2R, not in the S1 conformation, whereas this water molecule would be present in the S1 conformation of Ach-LY2119620-bound of M2R, but not in the related S2 conformation? Is that right? This is confusing. Can it be written more clearly?

e) In the figure 7 panel b, an intracellular surface of M2R is shown. Accordingly, ICL1 and ICL2 should be indicated instead of ECL1 and ECL2. Moreover, the HSQC spectrum of the M2Rmini-delta5M apo form versus LY2119620-bound form is shown two times (panels c and d). This is not really necessary.

f) Why is there a signal from arrestin in panel i of extended data figure 6? Only the M2RFL-delta8M has been labeled with ¹³CH₃-methionines in these experiments, not the arrestin ? How explain that a signal from arrestin is recorded? It would be important to add a few sentences to explain this phenomenon.

g) Many different constructs of M2R have been designed, expressed, labeled, and/or purified in this study (a cryoEM version with ICL3 truncated, a M2Rmini-delta5M, a M2RFL-delta8M, a M2RFL-delta9M, a M2RFL-delta9M-Met383). Although there is a snake plot of M2R (M2RFL-delta8M) shown in panel a of extended data figure 6, it would be very helpful to have a comparative view of these different receptors into a single panel or even a novel extended figure to be able to directly visualize what version is used in each series of experiments. As it, it is not obvious.

We thank all the reviewers for their constructive and helpful comments. Please see our detailed responses to the comments below. The reviewers' comments are in black font and our responses are in blue font.

Reviewer #1 (Remarks to the Author)

This is an outstanding and comprehensive study, which delves into the structure-function relationship regarding the endogenous agonist of the GPCR, M2R, and a supramolecular agonist. Importantly, comparative cryo-EM studies were performed in complex with G protein both with GDP and after removal of nucleotide. The functional studies and NMR studies, the G protein activation assays and arrestin-1 assays all come together beautifully to connect structure, conformational heterogeneity, and function. Notably, with endogenous agonist the authors observe two very distinct activation states via cryo-EM, which they argue are both fully active signatures. This is not the case with the synthetic super-agonist and there are corresponding observations of the toggle switch and allosteric response at the alpha-5 and alpha-N helices, alongside corroborating ¹³Cmethyl TROSY signatures at 6.31 confirming what cryo-EM identifies. The authors then go on to identify mechanisms of a PAM which biases arresting association. This is important because prior X-ray structures revealed little conformational changes. Both NMR and cryo-EM identify signatures that could explain how this PAM achieves arrestin bias, although this data is slightly more speculative.

The paper is appropriately long and detailed and the comparisons to other muscarinic and NTS1 structures are appropriate and interesting. I think this paper should be published with only very minor corrections. Some grammar corrections are summarized in my attached pdf. (Anything highlighted in pink is awkward grammatically).

Thank the reviewer for the positive comments. We have checked the highlighted text for errors in grammar.

Reviewer #2 (Remarks to the Author)

Xu et al. determined the cryo-EM structures of M2R-GoA complexes in the presence of the endogenous agonist acetylcholine or a supra-physiological agonist iperexo with or without an allosteric modulator LY2119620 bound. These structures reveal that iperexo stabilizes a uniform M2R-GoA complex, while acetylcholine stabilizes a more heterogeneous complex. Based on the biochemical and signaling

assays, LY2119620 seems to be an arrestin-biased allosteric modulator that enhances arrestin recruitment while impairing G protein activation. Structural and NMR spectroscopic data provide interesting insights into mechanisms underlying super-agonism of iperoxo and biased signaling of LY2119620. The manuscript is well written and illustrated.

We thank the reviewer for the positive comments.

However, I have a few comments that need to be addressed before I can recommend it for publication in Nature Communications:

Major points:

1. For monitoring Gi/o activation by M2R, the authors performed G-protein IP-one accumulation assay with a chimera Gaqi where the last five amino acids at the C-terminus of Gaq are replaced by the equivalent residues of Gai1. However, the structures of M2R-GoA complex show that in addition to the C-terminal helix ($\alpha 5$) of GaoA, the other region of GaoA is involved in binding M2R. In addition, the last five residues of Gao are not identical to those of Gai. I am a bit worried whether Gaqi can recapitulate Gai/o in vivo?

Thank the reviewer for raising this point. We agree with the reviewer that the engineered Gaqi might not recapitulate the Gai/o function in vivo.

We actually discussed this possibility in our manuscript on page 5

“**Because the engineered G-proteins used in these assays or the endogenous G-proteins in cell membrane could alter the functional outcomes...**” Therefore, in this study, we also performed in vitro functional assays using purified WT GoA protein to further confirm the key functional outcomes of these ligands that were observed in the G-protein IP-one assay (e.g. the super efficacy of iperoxo and the negative allosteric effects of LY2119620).

2. Interpretation of bias signaling should take account into both the potency (EC50) and efficacy (Emax). LY2119620 increases the potencies (EC50) of both Ach and Ixo in the G-protein IP-One assay to a greater extent than those in the arrestin-recruitment assay. Besides, only high concentrations of LY2119620 (30 μ M, 100 μ M) can decrease the efficacies of Ach and Ixo in the IP-One assay, whereas relatively low concentrations of LY2119620 (0.1-10 μ M) show minimal effects on their efficacies while significantly increasing their potencies. These signaling assay results suggest that LY2119620 enhances both the G protein activation and arrestin recruitment. Thus, the claim that LY2119620 impairs G protein activation may be not accurate. I suggest the authors tone down the arrestin-biased activity of LY2119620.

We thank the reviewer for the thoughtful comments. Indeed, in the G-protein IP-one assay, only high concentrations of LY2119620 show decreased G-protein efficacy. While this inhibitory effect of LY2119620 was confirmed by the in vitro GTPase-Glo assay. Therefore, we think the LY2119620 is an allosteric ligand that can increase ligand potency but reduce G-protein efficacy. Such allosteric ligands have also been reported for several other GPCRs, for example, the CB1 allosteric modulator org27569 (Jonathan F. Fay and David L. Farrens, 2015 PNAS).

The reviewer is correct, that signaling bias involves both potency and efficacy. We have toned down the discussion regarding the arrestin-biased activity of LY2119620 by rephrasing the claim in several places in our revised manuscript, including the title and abstract, and just specifically claim that LY2119620 can reduce G-protein efficacy but enhance arrestin efficacy.

3. It is interesting to obtain two different conformational states of the Ach-bound M2R-GoA complex with two distinct G-protein orientations, which were also observed in the NTSR1-Gi complex. Both M2R and NTSR1 contain a large hydrophobic residue at the position 34.51 of ICL2 (L in M2R, F in NTSR1). The hydrophobic interactions between the residue 34.51 and G α are stronger in one state than the other state. In fact, most structures of GPCR-Gi/o complex lack these strong hydrophobic interactions. So whether strong hydrophobic interactions between the residue 34.51 and G α i/o observed in one state (S1 state in M2R) is essential for Gi/o coupling of M2R. Mutagenesis studies are highly encouraged to demonstrate the functional relevance of the two states.

In the revised manuscript, we included the mutagenesis data of L129^{34.51}A (Rebuttal_Fig.1). The results show that the L129A mutant can still couple with G-protein but with significantly reduced activity for both ACh and Ixo. These data suggest that the strong interactions between residue 34.51 and Gi/o are important for the full activity of Gi/o coupling. This figure has been put into a supplementary figure in the revised manuscript (Extended Data Fig. 5). We have added discussion on page 10 as follow:

“The hydrophobic contacts are much stronger in the S1 state than in the S2 state, mainly mediated by the conserved ICL2 residue L129^{34.51} (Fig. 4e-g). Replacement of L129^{34.51} with alanine can significantly reduce the pEC50 for both ACh and Ixo, suggesting the important role of this residue in the full activity of GoA coupling (Extended Data Fig. 5).”

Rebuttal-Fig. 1 Mutagenesis and function data of L129^{34.51A}

4. Related to Fig. 5e-5g, the concentrations of ACh and Ixo should be provided in the method section or Figure legends. If sub-stoichiometric amount of ligands are added, the two NMR peaks of M383 in M2R when bound to ACh or Ixo should represent the ligand-bound state and apo-state which exchange slowly. If the receptor is saturated by the ligand, the two peaks may represent two different conformational states of M2R when bound to the ligand. The description in the fifth line of PAGE 12 should be revised accordingly.

Thank the reviewer for pointing this out. We do use saturating concentrations of ligands (1mM for both agonists) in our NMR experiments to make sure all of the receptor is ligand-bound. We have added such information (Page 34) in the methods part of the revised manuscript.

“Both agonists were added to the NMR sample at concentrations of at least 10-fold stoichiometric excess over the receptor (around 1mM), and more than 10-fold over their K_i values to ensure that the receptors were fully occupied by the ligands”.

Also, we have revised the fifth line of page 12 as follows:

“When bound to ACh or Ixo, similar spectral changes were observed, where two different conformational states were detected”

Minor points:

1. Every data point needs to be provided in graphs, Fig 1m, Fig 1i-l.

Data points were added.

2. The 9th line in PAGE 8, W4227.53 should be W4227.35.

We have corrected this.

3. BW numbers need to be added in Fig. 4.

We have added the BW numbers.

4. PAGE 17, miniGo α should be miniG α .

We have corrected miniGo α with miniG α in all the manuscript.

5. Fig. 2b, the map of G γ is not shown.

We have revised the figure and now the G γ is shown.

6. The outlier residues in Ramachandran plot need to be fixed.

We have fixed all the Ramachandran outliers for the models of S2 state, as shown by the new PDB validation reports.

Rebuttal-Fig. 2 Overall quality of model of ACh_S2 state

Rebuttal-Fig. 3 Overall quality of model of LY2119620-bound ACh_S2 state

7. EC50 should be changed to EC50 for consistency.

We have corrected this.

Reviewer #3 (Remarks to the Author)

The manuscript by Xu J. et al. describes structural and dynamic insights into the activation and biased allosteric modulation of the M2 muscarinic acetylcholine receptor. The combination of cryo-electron microscopy (cryoEM) and NMR spectroscopy allows to better understand regulation of this prototypical G protein-coupled receptor (GPCR) by acetylcholine and iperoxo agonists in the presence of the arrestin-biased allosteric modulator LY2119620.

This work is an outstanding study. While both the cryoEM structures and the previous crystal structures with or without LY2119620 show nearly identical intracellular conformations, the NMR spectroscopy reveals in details that the allosteric modulator bound together with an agonist can stabilize distinct active conformations in the intracellular domain compared to those bound to agonist alone. It shows how different structural approaches can be complementary to each other allowing to determine structural snapshots of agonist (+ allosteric modulator)-GPCR-G protein complexes with high resolution together with dynamic information (thanks to the labeling of methionine residues at crucial positions) in different receptor domains. In the future, this structural information will certainly help in the rational design and development of novel therapeutic ligands targeting muscarinic acetylcholine receptors. The expertise of the authors in the scientific field of GPCRs and structural biology is internationally well renowned, the approaches developed are state-of-the-art and the results presented in the article are convincing and very nicely illustrated. In particular, the sensor positions (labeled methionine) in the M2R are perfectly chosen and the different NMR HSQC spectra are quite clear and unambiguous.

Thank the reviewer for the positive comments on our work.

I have no major criticisms; I only have some minor comments.

a) In the Methods section, more precisely in the radioligand binding assay paragraph (page 37), would it be possible to explain how the HDL-reconstituted M2 receptor is prepared? At least, please give a reference or briefly detail the protocol. Only the radioligand binding assay is described, not the purification of the HDL-inserted M2R particle.

In the revised manuscript, we have added the reference for HDL reconstitution in the Methods section (Page 37) as follows:

***“Purified M2R was reconstituted into high-density lipoprotein (HDL) particles constituting apolipoprotein A1(APOA1) and a mixture of***

POPG: POPC lipids with 3:2 (mol:mol)ratio as previously described (ref, Whorton et al, 2007)."

b) In the extended data figure 3 panel e-h, it would be important to have an idea of the LY2119620 orientation in the extracellular vestibule. In other words, what are the ECLs seen in these close-up views (probably ECL2 and 3)? Please indicate ECL2 and ECL3, like in the panel a of figure 6.

We have labeled the ECL2 and ECL3 in the revised figure.

c) The NMR resonance of K383M is assigned at 2.3 ppm and 17.2 ppm in the ¹³C and ¹H dimension respectively (main text page 12 and extended data figure 4e). This is a mistake. Indeed, it is rather 2.16 ppm and 17.2 ppm in the ¹H and the ¹³C dimension, respectively. Please correct.

We have corrected this.

d) Is there a contradiction between figure 6 panel d and extended data figure 5 panel b? If well understood, a density for a water molecule is probably seen in the cryoEM-determined S2 conformation of Ach-bound M2R, not in the S1 conformation, whereas this water molecule would be present in the S1 conformation of Ach-LY2119620-bound of M2R, but not in the related S2 conformation? Is that right? This is confusing. Can it be written more clearly?

Sorry for the confusion. There is actually no contradiction. In our cryo-EM maps, we only observe extra electron density for the water molecule in the S2 conformation of ACh-bound map, and in S1 conformation of ACh+LY2119620 bound map. We think the water molecule exists in all conformations, however, the density was not well resolved in two of the maps. To make this clearer, we have rephrased the description in Page 14 as follow:

"The water molecule mediated polar interactions between ACh and N404^{6.52} are found in the S1 state only when LY2119620 is bound, and in the S2 state only in the absence of LY2119620 (Fig. 3f and Extended Data Fig. 6a.). It is likely that the water-mediated interaction exists in all conformations; however, the density was not well resolved in two of the maps."

e) In the figure 7 panel b, an intracellular surface of M2R is shown. Accordingly, ICL1 and ICL2 should be indicated instead of ECL1 and ECL2. Moreover, the HSQC spectrum of the M2Rmini-delta5M apo form

versus LY2119620-bound form is shown two times (panels c and d). This is not really necessary.

We have corrected the figure and removed the redundant spectrum.

f) Why is there a signal from arrestin in panel i of extended data figure 6? Only the M2RFL-delta8M has been labeled with ^{13}C -methionines in these experiments, not the arrestin? How explain that a signal from arrestin is recorded? It would be important to add a few sentences to explain this phenomenon.

The strong signal indicated in panel i of extended data figure 6 only comes up when adding arrestin. We think the signal is likely originated from the natural abundant ^{13}C (most likely methionine or alanine in this region) in arrestin. Because arrestin is a relatively small soluble protein and we added it in high concentration, so even relatively low abundant natural ^{13}C will give a strong signal. We collected a spectrum of arrestin alone, which also shows same signal at around 2.1 ppm in ^1H dimension and 17 ppm in ^{13}C dimension (See below). We have added a description in the Figure legend of extended data figure 6 :

“The asterisk indicates natural abundant signals from βArr1 , which only appear in the presence of βArr1 ”

Rebuttal-Fig. 4 Spectrum of arrestin

g) Many different constructs of M2R have been designed, expressed, labeled, and/or purified in this study (a cryoEM version with ICL3 truncated, a M2Rmini-delta5M, a M2RFL-delta8M, a M2RFL-delta9M, a M2RFL-delta9M-Met383). Although there is a snake plot of M2R (M2RFL-delta8M) shown in panel a of extended data figure 6, it would be very helpful to have a comparative view of these different receptors into a single panel or even a novel extended figure to be able to directly

visualize what version is used in each series of experiments. As it, it is not obvious.

We have made a new snake plot of M2R where all constructs used in this study are illustrated together for better comparison. The new figure was incorporated in **Extended data Fig1c** now. The Extended data Fig. 6 and all figure legends were revised accordingly.

REVIEWERS' COMMENTS

Reviewer #2 (Remarks to the Author):

The authors adequately address my concerns. I support its publication.

Reviewer #2 (Remarks to the Author):

The authors adequately address my concerns. I support its publication.

We thank the reviewer's constructive suggestions and support for the publication of our manuscript.